# Stochastic Order Learning: An Approach to Rank Estimation Using Noisy Data

Chaewon Lee [1]  Seon-Ho Lee [2]  Chang-Su Kim [1]

## Abstract

Rank estimation under label noise poses a fundamental challenge, as ordinal annotations often exhibit structured uncertainty rather than simple label corruption. In this paper, we reformulate rank estimation with noisy ordinal labels as a stochastic ordering problem, in which each instance is inherently associated with multiple plausible ranks instead of a single deterministic label. Based on this view, we propose stochastic order learning (SOL), a learning framework that captures ordinal label uncertainty and learns an embedding space through two complementary objectives: a discriminative loss that structures instance–centroid interactions and a stochastic order loss that enforces probabilistic ordering relations between instances. Extensive experiments across diverse datasets demonstrate that SOL enables reliable rank estimation under various types and levels of label noise. The source code is available at https://github.com/cwlee00/SOL.

## 1. Introduction

Rank estimation — a task to predict the rank or 'ordered class' of an object — is a fundamental problem in machine learning, with applications including facial age estimation (Ricanek & Tesafaye, 2006; Shin et al., 2022), aesthetic score regression (Kong et al., 2016), and medical assessment (Halabi et al., 2019). In practice, however, obtaining error-free ordinal annotations is challenging, as distinctions between adjacent labels are often subtle. For instance, facial appearance changes little over short age gaps, making annotation errors inevitable; indeed, Escalera et al. (2015) showed that apparent age distributions differ from real ages. Label noise also arises from subjectivity, as in aesthetic assessment where no universal scoring criterion exists, and from inter-observer variability in medical image analysis

(Halabi et al., 2019). To mitigate such variability, annotations are often aggregated by averaging estimates from multiple experts.

Many algorithms have been developed to train machines using imperfect data with noisy labels, but most of them are for classification (Tanno et al., 2019; Song et al., 2019; Ma et al., 2020; Yao et al., 2022; Ye et al., 2023) or segmentation (Yang et al., 2020; Li et al., 2023). Unlike classification or segmentation, rank estimation suffers from varying degrees of label errors due to the ordinal property of classes. Figure 1 compares nominal data for classification and ordered data for rank estimation. In classification, misclassifying a dog as a cat is as harmful as misclassifying a dog as a bear. In contrast, in rank estimation, the error of estimating a 43-year-old as a 59-year-old is severer than that of mistaking a 24-year-old as a 26-year-old. Since noise-robust classification methods treat all noise identically, they are prone to making big estimation errors and are incapable of identifying extreme outliers when applied to ordered data.

Although several noise-robust regression methods exist, regression-based models are known to underperform compared to classification- or ranking-based methods. As pointed out by Zhang et al. (2023), direct regression may fail to learn high-entropy feature representations, resulting in lower mutual information between learned representations and target outputs. Order learning approaches (Lim et al., 2020; Shin et al., 2022; Lee et al., 2022) overcome the limitations of direct regression and have shown promising results in rank estimation. However, these methods assume clean annotations, and their performance degrades in the presence of label noise, highlighting the need for noise-robust order learning algorithms.

Importantly, learning from noisy ordinal labels is not merely a straightforward extension of order learning with additional robustness mechanisms. Existing order learning methods rely on a deterministic association between each instance and a single, well-defined rank—an assumption that breaks down once ordinal labels are corrupted by noise. In such settings, an instance no longer corresponds to a unique rank but instead relates to multiple neighboring ranks with varying degrees of plausibility, reflecting both the severity and the structure of ordinal label errors. This form of structural uncertainty cannot be adequately captured by treating noise

---

[1]School of Electrical Engineering, Korea University, Seoul, Korea [2]Amazon AGI, Seattle, USA. Correspondence to: Chang-Su Kim <changsukim@korea.ac.kr>.

*Proceedings of the 43rd International Conference on Machine Learning*, Seoul, South Korea. PMLR 306, 2026. Copyright 2026 by the author(s).

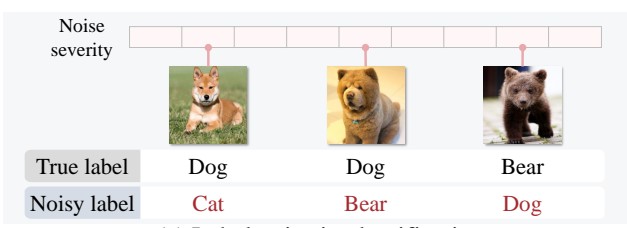

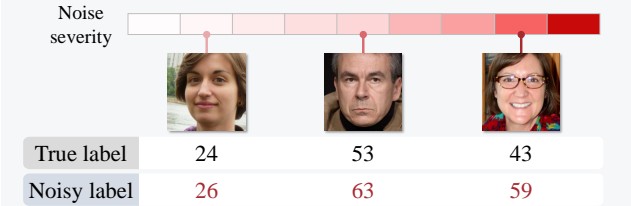

(a) Label noise in classification        (b) Label noise in rank estimation

*Figure 1.* Nominal data in classification versus ordered data in rank estimation. Unlike classification, in rank estimation, certain errors are severer than others.

as a marginal perturbation to otherwise deterministic supervision. Rather, rank estimation under label noise must be reformulated as a stochastic ordering problem, in which instance–rank relationships are inherently probabilistic rather than fixed.

In this paper, we propose a learning framework, called stochastic order learning (SOL), that instantiates the aforementioned stochastic ordering view of rank estimation under label noise. Given noisy ordinal annotations, SOL captures label uncertainty by associating each instance with a set of plausible ranks rather than a single deterministic label. Based on this probabilistic instance–rank association, we learn an embedding space guided by a desideratum that minimizes dissimilarities between instances and their corresponding rank centroids in expectation. This formulation naturally yields two complementary objectives: a discriminative loss that governs instance–centroid attraction and repulsion, and a stochastic order loss that enforces ordering relations between instances. In addition, SOL can optionally incorporate an outlier identification and relabeling step to refine particularly noisy instances. Extensive experiments show that SOL enables reliable rank estimation across diverse ordered datasets under various types and levels of label noise.

The contributions of this paper can be summarized as follows.

- We reformulate rank estimation under label noise as a stochastic ordering problem, in which the relationship between an instance and its rank is inherently probabilistic rather than deterministic.
- Based on this reformulation, we propose stochastic order learning (SOL), a learning framework that models ordinal label uncertainty and learns an embedding space through two complementary objectives: a discriminative loss for instance–centroid interactions and a stochastic order loss for probabilistic ordering relations.
- We conduct extensive experiments across diverse benchmarks in vision, medical imaging, aesthetics assessment, and natural language processing, demonstrating that SOL enables reliable rank estimation under various types and levels of label noise.

## 2. Related Work

**Learning from noisy labels:** Early work on learning from noisy supervision focused on correcting inconsistent annotations by enforcing structural constraints, such as isotonic classification and rule learning with monotonicity constraints (Chandrasekaran et al., 2005; Kotłowski & Słowiński, 2009). With the rise of deep learning and large-scale datasets, label noise has become a more prominent challenge, as deep neural networks are particularly sensitive to corrupted supervision. Consequently, numerous approaches have been proposed, including robust loss functions (Ghosh et al., 2017; Zhang & Sabuncu, 2018; Lyu & Tsang, 2019; Ma et al., 2020; Ye et al., 2023), noise-tolerant objectives such as peer loss (Liu & Guo, 2020), regularization strategies (Tanno et al., 2019; Menon et al., 2020; Xia et al., 2020), robust architectures (Han et al., 2018a; Goldberger & Ben-Reuven, 2022), selective data sampling (Han et al., 2018b; Jiang et al., 2018; Song et al., 2019), and representation-learning methods including selective supervised contrastive learning (Li et al., 2022b). However, the majority of these methods are designed for classification or segmentation tasks, rather than ordinal rank estimation.

Compared to classification, relatively few methods have addressed regression and ordinal prediction under label noise. Classical ordinal regression models, such as cumulative link and stereotype models (McCullagh, 1980; Anderson, 1984), handle ordered labels through latent-variable likelihood formulations, but are restricted to parametric settings and do not support representation learning from high-dimensional inputs such as images or text. These limitations have motivated recent efforts to revisit ordinal regression under label noise within modern learning frameworks.

Inspired by the unbiased risk estimation framework of Natarajan et al. (2013), Garg & Manwani (2020) studied noise-robust ordinal regression by modifying the loss function so that optimization under corrupted labels is equivalent to that under clean labels. Castells et al. (2020) proposed to down-weight samples with large losses during training, based on the assumption that noisy samples tend to incur higher losses. Yao et al. (2022) adapted Mixup (Zhang et al., 2018) for regression by sampling training pairs with

*Figure 2.* Overview of the proposed SOL algorithm

closer ordinal labels more frequently. Wang et al. (2022a) showed that standard regularization schemes are ineffective under label noise and proposed a noise-robust text regression method that mitigates noise by discarding or repairing detected noisy samples. More recently, Kim et al. (2024) introduced a contrastive fragmentation strategy that partitions the label space into fragments and trains expert extractors on fragment pairs for robust feature learning, while leveraging neighborhood agreement to identify clean samples.

**Rank estimation:** Rank estimation aims to predict the ordered class of an object and differs fundamentally from ordinary classification. Early approaches estimated ranks using either direct regression or classification models. Direct regression (Guo et al., 2009), which predicts scalar values, often performs poorly because it disregards the underlying physical or semantic processes associated with ranks, such as aging. Classification-based methods (Geng et al., 2007) formulate rank estimation as a multi-class classification problem, but fail to explicitly model the ordinal relationships among labels. To address this limitation, ordinal regression methods decompose rank estimation into a sequence of binary classification sub-problems (Frank & Hall, 2001; Li & Lin, 2006). More recently, deep ordinal regression techniques have been proposed, including pairwise regularization (Liu et al., 2018), soft labels (Diaz & Marathe, 2019), continuity-aware probabilistic networks (Li et al., 2019), and uncertainty-aware regression (Li et al., 2021). Related to ambiguity modeling, Gao et al. (2017) represented ordinal labels using Gaussian distributions to capture proximity-induced ambiguity, but did not account for noise arising from incorrect annotations.

**Order learning:** Order learning (Lim et al., 2020) is an approach to rank estimation motivated by the observation that relative assessment is often easier than absolute prediction. Instead of directly predicting ranks, Lim et al. (2020) estimate the rank of an instance by comparing it against reference instances with known ranks. To improve reference reliability, Lee & Kim (2021) proposed order–identity decomposition. Subsequent work extended order learning to regression settings (Shin et al., 2022), as well as weakly supervised and unsupervised scenarios (Lee & Kim, 2022; Lee et al., 2024). More recently, Lee et al. (2022) exploited both ordering relations and metric information by learning an embedding space in which instances are arranged according to their ranks. However, existing order learning methods assume deterministic and error-free rank annotations, and therefore do not model uncertainty arising from noisy ordinal labels.

## 3. Proposed Algorithm

### 3.1. Problem Formulation

There is a training set $\mathcal{X}$, in which each instance is attributed with one of the $n$ ranks (or ordered classes), represented by consecutive integers in $\{1, \ldots, n\}$. Let $\bar{r}_x$ denote the true rank of instance $x \in \mathcal{X}$. However, only a noisy rank $r_x$ is available, given by

$$r_x = \bar{r}_x + e_x \qquad (1)$$

where $e_x$ is the label error of $x$. Let $\mathbf{e}$ be the random variable underlying each error $e_x$. It is assumed that $\mathbf{e}$ has a discrete Gaussian distribution;

$$p_s \triangleq \Pr(\mathbf{e} = s) = \frac{1}{C} e^{-\frac{s^2}{2\sigma^2}} \qquad (2)$$

where $C = \sum_t e^{-\frac{t^2}{2\sigma^2}}$, and $s, t \in \mathbb{Z}$. Note that the noise distribution in (2) is symmetric ($p_s = p_{-s}$) and unimodal ($p_s \geq p_t$ for $0 \leq s \leq t$). This provides a simple and practical model for label errors in many real-world rank estimation tasks. For example, it is more likely for an annotator to mislabel a 10-year-old as 8 or 12 years old than as 20 years old. While we adopt a discrete Gaussian model for clarity and tractability, our approach does not rely on this specific distribution, and its robustness under alternative noise types is evaluated empirically in Section 4.

We employ an encoder $h$ to map each instance $x \in \mathcal{X}$ into a feature vector $h_x = h(x)$ in an embedding space, as shown in Figure 2. We aim to construct the embedding space in which the instances are arranged according to their ranks, and each 'centroid' $\mu_r$ is the representative vector for instances with rank $r \in \{1, \ldots, n\}$. However, since only the noisy rank $r_x$ in (1) — instead of the true rank $\bar{r}_x$ — is available, instance $x$ relates stochastically to multiple centroids, rather than deterministically to the single centroid

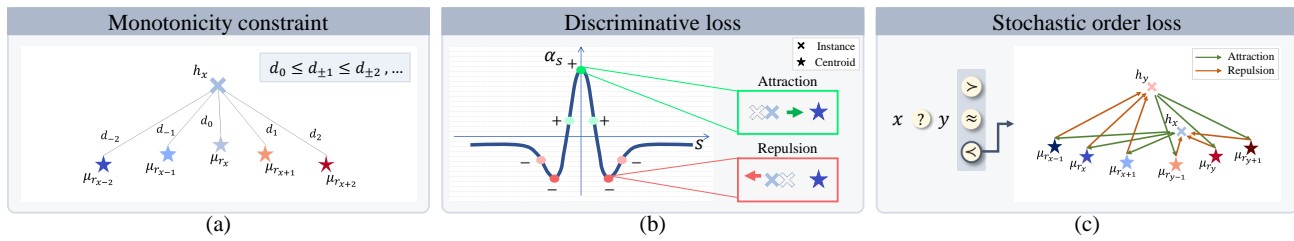

Figure 3. Illustration of the monotonicity constraint and the training losses for constructing a SOL embedding space

$\mu_{\bar{r}_x}$. Specifically, $x$ is associated with $\mu_{r_x-s}$ with probability $p_s$ in (2). Note that, due to the symmetry $p_s = p_{-s}$, $x$ is also associated with $\mu_{r_x+s}$ with $p_s$. Thus, in the embedding space, the mean squared distance $\sum_s p_s d^2(h_x, \mu_{r_x+s})$ should be minimized, where $d$ denotes the Euclidean distance and the summation is taken over valid ranks.

We hence define the stochastic dissimilarity of instance $x$ from rank $r$ in the embedding space determined by the encoder $h$ as

$$D_h(x, r) = \sum_s p_s d^2(h_x, \mu_{r+s}). \quad (3)$$

Then, the objective of SOL is to design the encoder $h$ satisfying the following desideratum for each $x \in \mathcal{X}$:

$$D_h(x, r_x) \leq D_h(x, r) \quad \text{for all } r \in \{1, \ldots, n\}. \quad (4)$$

A sufficient condition for satisfying this desideratum is the monotonicity constraint, given by

$$d(h_x, \mu_{r_x+s}) \leq d(h_x, \mu_{r_x+t}) \text{ for all } |s| \leq |t|, \quad (5)$$

as proven in Appendix A. Intuitively speaking, this monotonicity can be achieved, provided that the centroids are arranged directionally according to the ranks, and the instance $h_x$ is located near the centroid $\mu_{r_x}$, as illustrated in Figure 3(a).

In the inference phase, based on the desideratum in (4), we estimate the rank of an unseen instance $x$ by

$$\hat{r}_x = \arg\min_{r \in \{1, \ldots, n\}} D_h(x, r). \quad (6)$$

### 3.2. Stochastic Order Learning

To learn or construct an embedding space in which instances and centroids are well aligned according to the desideratum in (4), we optimize the parameters of the encoder $h$ by minimizing the loss function

$$\ell_{\text{total}} = \sum_{x \in \mathcal{X}} \ell_{\text{disc}}(x) + \sum_{x,y \in \mathcal{X}} \ell_{\text{order}}(x, y) \quad (7)$$

where $\ell_{\text{disc}}$ is the discriminative loss, and $\ell_{\text{order}}$ is the stochastic order loss.

**Discriminative loss:** To encourage the desideratum in (4),

we employ the discriminative loss

$$\ell_{\text{disc}}(x) = \sum_{t=1}^{T} \big( D_h(x, r_x) - D_h(x, r_x + t)$$
$$+ D_h(x, r_x) - D_h(x, r_x - t)\big) \quad (8)$$
$$= \sum_{t=1}^{T} \sum_s (2p_s - p_{s-t} - p_{s+t}) d^2(h_x, \mu_{r_x+s}) \quad (9)$$
$$= \sum_s \alpha_s d^2(h_x, \mu_{r_x+s}) \quad (10)$$

where $\alpha_s = \sum_{t=1}^{T} (2p_s - p_{s-t} - p_{s+t})$. Also, $T$ is a hyperparameter, and its impacts are analyzed in Appendix D.1. Note that each difference in (8) is non-positive when the desideratum in (4) holds. Therefore, the discriminative loss is constructed as a direct surrogate of the desideratum, and minimizing it encourages the embedding to satisfy the desideratum.

Note that the coefficient $\alpha_s$ can be viewed as a discrete approximation of the second-order derivative of the Gaussian distribution, which exhibits inflection points. Therefore, there exists a threshold $\delta$ such that $\alpha_s$ is positive for $|s| < \delta$ and negative otherwise, as illustrated in Figure 3(b). As a result, minimizing the discriminative loss effectively reduces $d(h_x, \mu_{r_x+s})$ for $|s| < \delta$. In other words, $h_x$ is encouraged to move closer to the centroids corresponding to ranks within the range $(r_x - \delta, r_x + \delta)$. Conversely, when $|s| > \delta$, the loss encourages increasing $d(h_x, \mu_{r_x+s})$, thereby repelling $h_x$ from centroids corresponding to ranks outside $(r_x - \delta, r_x + \delta)$. To summarize, $\ell_{\text{disc}}$ attracts each $h_x$ toward the centroid $\mu_{r_x}$ and its neighboring centroids to account for label noise, while repelling it from other centroids.

**Stochastic order loss**: In order learning (Lim et al., 2020; Lee & Kim, 2021; Lee et al., 2022), pairwise relationships between instances are used to construct a desired embedding space. Thus, while the discriminative loss $\ell_{\text{disc}}$ in (8) considers the geometric configuration of a single instance $x$ with respect to the centroids, the stochastic order loss $\ell_{\text{order}}$ takes into account the joint geometric configuration of two instances $x$ and $y$.

There are three ordering cases between $x$ and $y$ (Lim et al., 2020):

$$
\begin{aligned}
x \prec y & \quad \text{if } \bar{r}_x - \bar{r}_y < -\tau, \\
x \approx y & \quad \text{if } |\bar{r}_x - \bar{r}_y| \leq \tau, \quad (11) \\
x \succ y & \quad \text{if } \bar{r}_x - \bar{r}_y > \tau
\end{aligned}
$$

where $\tau$ is a threshold. For these three cases, Lee et al. (2022) use margin losses to align instances according to the ranks. Similarly, the proposed $\ell_{\text{order}}$ is based on margin losses. But, unlike Lee et al. (2022), true ranks $\bar{r}_x$ and $\bar{r}_y$ are unknown in SOL. Also, each instance relates to multiple centroids randomly in SOL. We hence develop $\ell_{\text{order}}$ to address these differences.

Since only noisy ranks $r_x$ and $r_y$ are available, the true ranks $\bar{r}_x$ and $\bar{r}_y$ in (11) can be expressed using (1). Let $s$ and $t$ denote the label noise of samples $x$ and $y$, respectively. Then, $\bar{r}_x - \bar{r}_y = r_x - r_y - s + t$. As we model label noise as stochastic variables, we can compute the probabilities for the three ordering cases using (2):

$$\Pr(x \prec y) = \sum_s \sum_{t:r_x-r_y-s+t<-\tau} p_s p_t, \quad (12)$$

$$\Pr(x \approx y) = \sum_s \sum_{t:|r_x-r_y-s+t|\leq\tau} p_s p_t, \quad (13)$$

$$\Pr(x \succ y) = \sum_s \sum_{t:r_x-r_y-s+t>\tau} p_s p_t. \quad (14)$$

Then, we define the margin loss for the case $x \prec y$ as

$$\ell_{x \prec y} = \sum_{r \leq r_x} \max\{D_h(x,r) - D_h(y,r) + \gamma, 0\} \\ + \sum_{r \geq r_y} \max\{D_h(y,r) - D_h(x,r) + \gamma, 0\} \quad (15)$$

where $\gamma$ is a margin. In order to minimize the first sum in (15), $D_h(x,r) - D_h(y,r) = \sum_s p_s(d^2(h_x, \mu_{r+s}) - d^2(h_y, \mu_{r+s}))$ is encouraged to be reduced for $r \leq r_x$. Accordingly, $h_x$ is encouraged to be close to $\mu_{r+s}$, while $h_y$ is encouraged to be farther from $\mu_{r+s}$. Note that this effect is pronounced only for small offsets $s$ due to the Gaussian weights $p_s$. Similarly, for $r \geq r_y$ and small $s$, the loss encourages $h_y$ to be close to $\mu_{r+s}$ while pushing $h_x$ away from $\mu_{r+s}$. Hence, $\ell_{x \prec y}$ facilitates the arrangement of instances and centroids in the embedding space, as illustrated in Figure 3(c). Note that the loss $\ell_{x \succ y}$ for the case $x \succ y$ is formulated symmetrically.

Also, when $x \approx y$, $h_x$ and $h_y$ should be close to each other. We hence define

$$\ell_{x \approx y} = \sum_{r \in \{1,\dots,n\}} \max(|D_h(x,r) - D_h(y,r)| - \gamma, 0). \quad (16)$$

Overall, we define the stochastic order loss as

$$\ell_{\text{order}}(x,y) = \Pr(x \succ y)\ell_{x \succ y} + \Pr(x \approx y)\ell_{x \approx y} \\ + \Pr(x \prec y)\ell_{x \prec y}. \quad (17)$$

**Centroid rule**: Moreover, we determine each centroid $\mu_r$ to minimize $\sum_{x \in \mathcal{X}} D_h(x, r_x)$ based on the desideratum in (4),

$$\mu_r = \frac{\sum_{x \in \mathcal{X}} p_{r-r_x} h_x}{\sum_{x \in \mathcal{X}} p_{r-r_x}}, \quad r \in \{1, \dots, n\}, \quad (18)$$

---

**Algorithm 1** Stochastic Order Learning (SOL)

**Input:** noisy dataset $\mathcal{X}$, number of ranks $n$, $\sigma$ in (2), $T$ in (8), $\tau$ in (11), $\beta$ in (19)
Initialize centroids $\{\mu_r\}_{r=1}^n$ via (18)
**repeat**
    Fine-tune encoder $h$ by minimizing $\ell_{\text{total}}$ in (7) {Network training}
    **for** $r = 1$ **to** $n$ **do**
        Update centroid $\mu_r$ via (18) {Centroid rule}
    **end for**
    **for all** $x \in \mathcal{X}$ **do**
        Estimate rank of $x$ via (6)
    **end for**
    Detect outliers $\bigcup_{r=1}^n \mathcal{X}_r$ via (19) {Outlier detection}
    **for all** $x \in \bigcup_{r=1}^n \mathcal{X}_r$ **do**
        Estimate label noise $\hat{e}_x$ via (20)
        Refine label of $x$ via (21) {Relabeling}
    **end for**
**until** predefined number of epochs
**Output:** updated labels $\{r_x\}$, centroids $\{\mu_r\}_{r=1}^n$, encoder $h$

---

as derived in Appendix B. We update the centroids after every training epoch.

### 3.3. Outlier Detection and Relabeling

To obtain a more reliable rank estimator, we identify outliers, likely to have extreme label errors, among instances in the noisy training set and refine their labels by estimating the errors. Then, in turn, we fine-tune the encoder or equivalently revamp the embedding space, so the instances are better arranged based on the refined rank information.

**Outlier detection:** We first estimate the rank of each training instance $x$ using the inference rule in (6). Then, for each rank $r \in \{1, \dots, n\}$, we detect the set $\mathcal{X}_r$ of outliers by

$$\mathcal{X}_r = \{x : r_x = r \text{ and } |r_x - \hat{r}_x| \geq \beta \cdot \max_{y:r_y=r} |r_y - \hat{r}_y|\} \quad (19)$$

where $\beta \in (0, 1)$ is a constant to control the precision of the outlier detection.

**Relabeling:** For each detected outlier $x \in \bigcup_{r=1}^n \mathcal{X}_r$, we estimate its label error as

$$\hat{e}_x = \begin{cases} \frac{1}{2|\mathcal{X}|} \sum_{y \in \mathcal{X}} |r_y - \hat{r}_y| & \text{if } r_x > \hat{r}_x, \\ -\frac{1}{2|\mathcal{X}|} \sum_{y \in \mathcal{X}} |r_y - \hat{r}_y| & \text{if } r_x < \hat{r}_x. \end{cases} \quad (20)$$

Then, from (1), we refine the rank of $x$ by

$$r_x \leftarrow r_x - \hat{e}_x. \quad (21)$$

We note that, in (20), $|\hat{e}_x|$ is determined as half of the mean absolute difference between noisy and estimated ranks over all training instances. It is to prevent drastic changes in

*Table 1.* Performance comparison on the MORPH II dataset.

| ALGORITHM | GAUSSIAN $\kappa = 0.2$ | | GAUSSIAN $\kappa = 0.3$ | | GAUSSIAN $\kappa = 0.4$ | | LAPLACIAN $\kappa = 0.3$ | | UNIFORM $\kappa = 0.3$ | | SKEWED $\kappa = 0.3$ | |
|---|---|---|---|---|---|---|---|---|---|---|---|---|
| | MAE(↓) | CS(↑) | MAE(↓) | CS(↑) | MAE(↓) | CS(↑) | MAE(↓) | CS(↑) | MAE(↓) | CS(↑) | MAE(↓) | CS(↑) |
| SPR (WANG ET AL., 2022B) | 8.446 | 41.71 | 8.881 | 34.79 | 9.239 | 36.89 | 8.577 | 39.89 | 8.254 | 40.53 | 8.980 | 38.07 |
| ACL (YE ET AL., 2023) | 9.017 | 36.75 | 9.492 | 35.61 | 9.314 | 35.74 | 8.873 | 35.87 | 8.849 | 35.95 | 9.613 | 35.93 |
| ROR-CE (GARG & MANWANI, 2020) | 2.859 | 86.79 | 3.018 | 86.79 | 3.170 | 82.60 | 3.058 | 84.97 | 2.827 | 87.34 | 3.663 | 77.69 |
| C-MIXUP (YAO ET AL., 2022) | 3.063 | 82.26 | 3.393 | 77.21 | 3.395 | 76.84 | 3.772 | 71.77 | 3.306 | 77.78 | 3.378 | 77.69 |
| CONFRAG (KIM ET AL., 2024) | 2.878 | 84.06 | 3.000 | 82.06 | 3.255 | 78.96 | 3.102 | 80.33 | 2.763 | 84.70 | 3.333 | 78.14 |
| POE (LI ET AL., 2021) | 2.989 | 82.88 | 3.093 | 80.33 | 3.253 | 79.23 | 3.332 | 77.50 | 2.908 | 83.61 | 3.389 | 75.59 |
| MWR (SHIN ET AL., 2022) | 2.570 | 90.07 | 2.693 | 89.25 | 2.851 | 87.16 | 2.854 | 86.61 | 2.529 | 90.71 | 3.327 | 80.42 |
| GOL (LEE ET AL., 2022) | 2.516 | 90.89 | 2.671 | 89.07 | 2.861 | 85.97 | 2.846 | 86.16 | 2.509 | 90.26 | 3.351 | 82.51 |
| CONR (KERAMATI ET AL., 2024) | 3.296 | 77.87 | 3.382 | 77.60 | 3.924 | 70.49 | 3.486 | 75.23 | 3.168 | 80.42 | 3.610 | 74.86 |
| CLOC (PITAWELA ET AL., 2025) | 4.461 | 67.58 | 5.966 | 52.64 | 6.232 | 50.27 | 8.018 | 38.43 | 3.558 | 79.87 | 3.502 | 79.14 |
| DHRL (SUZUKI ET AL., 2026) | 2.609 | 70.49 | 2.737 | 85.34 | 2.870 | 82.60 | 2.992 | 80.41 | 2.617 | 86.70 | 3.305 | 78.05 |
| SOL | **2.489** | **91.35** | **2.663** | **89.62** | **2.826** | **87.70** | **2.794** | **86.89** | **2.499** | **90.89** | **3.296** | **83.15** |

|  | | | | | | |
|---|---|---|---|---|---|---|
| Input image | | | | | | |
| True label | 23 | 25 | 42 | | 23 | 40 |
| SPR | 36 (+13) | 43 (+18) | 36 (−6) | | 48 (+25) | 25 (−15) |
| GOL | 27 (+4) | 20 (−5) | 46 (+4) | | 40 (+17) | 29 (−11) |
| SOL | 23 (+0) | 25 (+0) | 42 (+0) | | 36 (+13) | 33 (−7) |
| | (a) | | | | (b) | |

*Figure 4.* (a) Success and (b) failure cases of age estimation results on the MORPH II dataset. Under each image, we compare the estimated ages of SPR (Wang et al., 2022b), GOL (Lee et al., 2022), and the proposed SOL and specify the corresponding errors inside the parentheses.

rank labels, which may rather increase the label errors after relabeling. We repeat the encoder fine-tuning and the outlier detection and relabeling alternately to gradually reduce the label errors and construct a better embedding space. Algorithm 1 summarizes the overall process of SOL.

# 4. Experimental Results

We evaluate SOL on multiple rank estimation benchmarks, including facial age estimation (MORPH II (Ricanek & Tesafaye, 2006), CLAP2015 (Escalera et al., 2015)), aesthetic score regression (AADB (Kong et al., 2016)), medical assessment (RSNA (Halabi et al., 2019)), and textual regression (WMT2020 (Specia et al., 2020)).

We assess robustness under both synthetic and real-world noisy settings. For synthetic noise, we corrupt training labels using zero-mean discrete Gaussian noise as in prior work (Yao et al., 2022; Kim et al., 2024), with standard deviation

$$\sigma = \kappa \cdot \sigma_{\mathcal{X}}, \qquad (22)$$

where $\kappa \in (0, 1)$ controls noise severity and $\sigma_{\mathcal{X}}$ denotes the standard deviation of ground-truth ranks. At test time, since the true noise level is unknown, we fix $\sigma_{\text{test}}$ when computing $p_s$ in (2), independent of $\kappa$. We further consider

Laplacian and uniform noise perturbations to evaluate robustness across different noise types. For real-world noise, we apply SOL to a textual regression task with inherently noisy human annotations.

Additional dataset and noise-generation details are provided in Appendix C.

## 4.1. Implementation

We adopt VGG16 (Simonyan & Zisserman, 2015), initialized with the pre-trained parameters on ILSVRC2012 (Deng et al., 2009), as the encoder $h$. We use the Adam optimizer (Kingma & Ba, 2015) with a batch size of 32 and a weight decay of $5 \times 10^{-4}$. For data augmentation, we do random horizontal flips and random crops. More implementation details including hyperparameter settings are available in Appendix C, and experimental analysis on the hyperparameters is performed in Appendix D.1.

## 4.2. Comparative Assessment

We compare the proposed SOL with recent noise-robust classification methods (Wang et al., 2022b; Ye et al., 2023), noise-robust regression methods (Garg & Manwani, 2020; Yao et al., 2022; Kim et al., 2024), and rank estimation methods (Li et al., 2021; Shin et al., 2022; Lee et al., 2022; Keramati et al., 2024; Pitawela et al., 2025; Suzuki et al., 2026). For a fair comparison, the same backbone of VGG16 (Simonyan & Zisserman, 2015) is used for all methods. For evaluation, we adopt the mean absolute error (MAE) and cumulative score (CS) metrics: MAE is the average absolute error between estimated and ground-truth ranks, and CS computes the percentage of instances whose absolute estimation errors are less than or equal to a tolerance value. The tolerance value is 5 for MORPH II and CLAP2015, 0.25 for AADB, and 12 for RSNA. Justification for the choice of tolerance values is in Appendix C.4.

**Age estimation:** For facial age estimation, we employ two popular datasets MORPH II and CLAP2015. Table 1 com-

*Table 2.* Performance comparison on the CLAP2015 dataset.

| | GAUSSIAN | | | | | | LAPLACIAN | | UNIFORM | | SKEWED | |
| | $\kappa = 0.2$ | | $\kappa = 0.3$ | | $\kappa = 0.4$ | | $\kappa = 0.3$ | | $\kappa = 0.3$ | | $\kappa = 0.3$ | |
| ALGORITHM | MAE(↓) | CS(↑) | MAE(↓) | CS(↑) | MAE(↓) | CS(↑) | MAE(↓) | CS(↑) | MAE(↓) | CS(↑) | MAE(↓) | CS(↑) |
|---|---|---|---|---|---|---|---|---|---|---|---|---|
| SPR (WANG ET AL., 2022B) | 9.170 | 44.21 | 9.215 | 43.19 | 9.534 | 40.12 | 9.191 | 38.37 | 9.269 | 43.19 | 9.309 | 45.69 |
| ACL (YE ET AL., 2023) | 9.483 | 41.06 | 9.239 | 39.57 | 9.583 | 45.23 | 9.312 | 42.69 | 9.742 | 44.81 | 9.388 | 45.25 |
| ROR-CE (GARG & MANWANI, 2020) | 4.163 | 72.85 | 4.432 | 70.06 | 4.900 | 66.27 | 4.789 | 67.19 | 4.174 | 74.42 | 4.650 | 69.42 |
| C-MIXUP (YAO ET AL., 2022) | 5.042 | 61.65 | 5.285 | 58.71 | 5.302 | 58.52 | 4.824 | 62.65 | 4.511 | 64.87 | 4.760 | 63.11 |
| CONFRAG (KIM ET AL., 2024) | 4.898 | 62.19 | 4.658 | 63.11 | 5.328 | 58.20 | 4.690 | 62.47 | 4.858 | 61.17 | 4.512 | 64.97 |
| POE (LI ET AL., 2021) | 4.052 | 70.34 | 4.169 | 68.86 | 4.390 | 65.52 | 4.303 | 66.64 | 4.061 | 69.32 | 4.401 | 64.97 |
| MWR (SHIN ET AL., 2022) | 3.577 | **79.80** | 3.830 | 76.18 | 4.299 | 72.85 | 4.011 | 74.05 | 3.685 | 77.39 | 4.415 | **70.06** |
| GOL (LEE ET AL., 2022) | 3.624 | 77.94 | 3.866 | 76.03 | 4.105 | 72.10 | 3.934 | 75.07 | 3.613 | 78.22 | 4.407 | 68.40 |
| CONR (KERAMATI ET AL., 2024) | 3.818 | 77.57 | 3.888 | 75.44 | 4.233 | 70.90 | 4.142 | 73.68 | 3.787 | 76.65 | 4.400 | 69.05 |
| CLOC (PITAWELA ET AL., 2025) | 5.594 | 58.39 | 7.098 | 51.44 | 7.165 | 48.66 | 8.001 | 46.15 | 6.596 | 54.96 | 6.140 | 57.65 |
| DHRL (SUZUKI ET AL., 2026) | 3.756 | 73.77 | 3.909 | 70.90 | 4.153 | 66.82 | 3.973 | 71.46 | 3.693 | 73.86 | 4.429 | 67.10 |
| SOL | **3.559** | 78.68 | **3.764** | **77.11** | **4.002** | **73.68** | **3.904** | **75.16** | **3.550** | **79.05** | **4.379** | 69.97 |

*Table 3.* Performance comparison on the AADB dataset.

| | GAUSSIAN | | | | | | LAPLACIAN | | UNIFORM | | SKEWED | |
| | $\kappa = 0.2$ | | $\kappa = 0.3$ | | $\kappa = 0.4$ | | $\kappa = 0.3$ | | $\kappa = 0.3$ | | $\kappa = 0.3$ | |
| ALGORITHM | MAE(↓) | CS(↑) | MAE(↓) | CS(↑) | MAE(↓) | CS(↑) | MAE(↓) | CS(↑) | MAE(↓) | CS(↑) | MAE(↓) | CS(↑) |
|---|---|---|---|---|---|---|---|---|---|---|---|---|
| SPR (WANG ET AL., 2022B) | 0.149 | 81.20 | 0.150 | 82.10 | 0.151 | 81.60 | 0.153 | 81.40 | 0.150 | 81.30 | 0.143 | 83.10 |
| ACL (YE ET AL., 2023) | 0.147 | 82.90 | 0.148 | 82.50 | 0.157 | 79.43 | 0.151 | 81.50 | 0.153 | 80.80 | 0.153 | 80.74 |
| ROR-CE (GARG & MANWANI, 2020) | 0.121 | 88.70 | 0.122 | 89.00 | 0.123 | 88.70 | 0.122 | 89.70 | 0.122 | 90.20 | 0.124 | 89.50 |
| C-MIXUP (YAO ET AL., 2022) | 0.119 | 91.13 | 0.122 | 89.31 | 0.130 | 88.51 | 0.121 | 90.50 | 0.121 | 90.90 | 0.123 | 90.70 |
| CONFRAG (KIM ET AL., 2024) | 0.129 | 88.00 | 0.126 | 88.70 | 0.134 | 86.90 | 0.126 | 89.00 | 0.124 | 89.70 | 0.123 | 88.60 |
| POE (LI ET AL., 2021) | 0.122 | 89.00 | 0.123 | 89.30 | 0.120 | 89.10 | 0.124 | 89.10 | 0.124 | 88.50 | 0.125 | 88.50 |
| MWR (SHIN ET AL., 2022) | 0.123 | 89.00 | 0.124 | 87.60 | 0.122 | 89.80 | 0.125 | 88.20 | 0.124 | 89.40 | 0.124 | 87.80 |
| GOL (LEE ET AL., 2022) | 0.114 | 92.40 | 0.117 | 91.80 | 0.119 | 91.00 | 0.118 | 91.50 | 0.117 | 91.60 | 0.120 | 91.00 |
| CONR (KERAMATI ET AL., 2024) | 0.117 | 91.60 | 0.116 | 91.70 | 0.118 | 91.00 | 0.118 | 91.30 | 0.118 | 91.90 | 0.119 | 91.20 |
| CLOC (PITAWELA ET AL., 2025) | 0.135 | 88.10 | 0.139 | 84.60 | 0.145 | 83.90 | 0.145 | 83.50 | 0.130 | 87.90 | 0.130 | 87.30 |
| DHRL (SUZUKI ET AL., 2026) | 0.131 | 87.30 | 0.134 | 87.00 | 0.135 | 86.60 | 0.134 | 86.80 | 0.132 | 87.50 | 0.136 | 87.00 |
| SOL | **0.111** | **92.70** | **0.114** | **93.20** | **0.115** | **92.00** | **0.115** | **92.30** | **0.116** | **93.30** | **0.118** | **92.30** |

pares the results on MORPH II. SPR (Wang et al., 2022b) and ACL (Ye et al., 2023), which are recent noise-robust classification methods, treat all label errors identically. Compared to rank estimation methods, they underperform because they fail to avoid making large estimation errors (*e.g.* absolute errors bigger than 20). The noise-robust regression methods ROR-CE (Garg & Manwani, 2020), C-Mixup (Yao et al., 2022), and ConFrag (Kim et al., 2024) perform better, for they penalize samples with severe errors. The rank estimation methods MWR (Shin et al., 2022) and GOL (Lee et al., 2022), along with more recent approaches such as ConR (Keramati et al., 2024), CLOC (Pitawela et al., 2025), and DHRL (Suzuki et al., 2026), provide stronger baselines. However, the proposed SOL outperforms all these methods without exception in terms of both MAE and CS.

We also provide examples of age estimation results in Figure 4. These examples are from MORPH II with Gaussian noise at $\kappa = 0.4$. We compare the prediction results on images for which SOL correctly estimates ages in Figure 4(a). Along with the successful cases, we also show some failure cases in Figure 4(b). Note that the noise-robust classifier SPR tends to make big errors as it fails to consider the ordinal property of age labels. The state-of-the-art rank esti-

mator GOL performs better with smaller errors. However, SOL manages to make closer estimates to the true ages than the other algorithms, in both successful and failure cases. Appendix D.14 presents more rank estimation results.

Table 2 lists the performances on CLAP. SOL again achieves the best MAE scores in all settings. Note that GOL also aims to sort instances according to their ranks in an embedding space. Compared to GOL, the proposed SOL provides better results in all cases, and the score gap generally gets bigger as the level of Gaussian noise ($\kappa$) increases. For example, the MAE score gap is 0.103 at $\kappa = 0.4$, while it is 0.065 at $\kappa = 0.2$. These results indicate that, despite label errors, SOL arranges the instances according to their true ranks more reliably. In other words, SOL is more noise-robust than GOL.

**Aesthetic score regression:** Table 3 compares the aesthetic score regression results on AADB. Since aesthetic assessment is inherently subjective and ambiguous, accurately predicting aesthetic scores is highly challenging. Nevertheless, the proposed SOL consistently achieves the best performance across all settings. At the highest Gaussian noise level $\kappa = 0.4$, SOL outperforms the second-best ConR by 2.5% in terms of MAE and improves the CS over the

*Table 4.* Performance comparison on the RSNA dataset.

| | GAUSSIAN | | | | | | LAPLACIAN | | UNIFORM | | SKEWED | |
| | $\kappa = 0.1$ | | $\kappa = 0.15$ | | $\kappa = 0.2$ | | $\kappa = 0.15$ | | $\kappa = 0.15$ | | $\kappa = 0.15$ | |
| ALGORITHM | MAE($\downarrow$) | CS($\uparrow$) | MAE($\downarrow$) | CS($\uparrow$) | MAE($\downarrow$) | CS($\uparrow$) | MAE($\downarrow$) | CS($\uparrow$) | MAE($\downarrow$) | CS($\uparrow$) | MAE($\downarrow$) | CS($\uparrow$) |
|---|---|---|---|---|---|---|---|---|---|---|---|---|
| SPR (WANG ET AL., 2022B) | 33.80 | 28.50 | 36.48 | 25.00 | 34.88 | 20.50 | 36.77 | 26.50 | 35.50 | 26.00 | 36.85 | 26.50 |
| ACL (YE ET AL., 2023) | 35.09 | 26.20 | 35.15 | 26.50 | 35.26 | 25.17 | 33.82 | 24.00 | 34.32 | 22.00 | 35.62 | 20.00 |
| ROR-CE (GARG & MANWANI, 2020) | 7.844 | 76.00 | 8.800 | 77.19 | 8.490 | 72.00 | 8.726 | 74.00 | 8.189 | 77.00 | 10.190 | 64.50 |
| C-MIXUP (YAO ET AL., 2022) | 8.200 | 72.40 | 8.621 | 69.71 | 9.054 | 66.70 | 10.603 | 62.00 | 10.124 | 67.00 | 10.504 | 67.00 |
| CONFRAG (KIM ET AL., 2024) | 8.287 | 76.50 | 8.458 | 77.50 | 8.805 | 71.50 | 8.977 | 74.50 | 8.995 | 73.00 | 8.814 | 72.00 |
| POE (LI ET AL., 2021) | 8.517 | 74.50 | 8.614 | 71.50 | 8.796 | 73.00 | 8.856 | 74.50 | 8.176 | 73.50 | 9.107 | 70.00 |
| MWR (SHIN ET AL., 2022) | _7.833_ | 75.00 | 8.239 | 77.50 | 8.353 | 72.00 | **8.272** | _76.00_ | 7.939 | 77.50 | _8.741_ | _72.50_ |
| GOL (LEE ET AL., 2022) | 8.170 | _77.50_ | _7.995_ | _80.00_ | _8.334_ | _75.00_ | 8.453 | 72.00 | _7.879_ | 77.50 | 8.994 | 71.00 |
| CONR (KERAMATI ET AL., 2024) | 8.239 | 72.50 | 8.449 | 74.50 | 9.017 | 74.50 | 9.078 | 72.50 | 8.844 | _78.00_ | 8.844 | 71.50 |
| CLOC (PITAWELA ET AL., 2025) | 9.721 | 71.50 | 9.890 | 69.00 | 10.250 | 64.50 | 9.807 | 66.00 | 9.346 | 72.50 | 9.722 | 72.00 |
| DHRL (SUZUKI ET AL., 2026) | 8.090 | 75.00 | 8.317 | 74.50 | 8.502 | 74.50 | 8.416 | 74.50 | 8.155 | _78.00_ | 8.788 | 70.00 |
| SOL | **7.579** | **78.50** | **7.706** | **80.50** | **8.051** | **76.50** | _8.289_ | **76.50** | **7.816** | **78.50** | **8.544** | **73.00** |

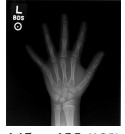 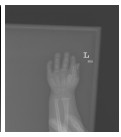 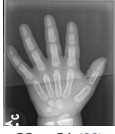 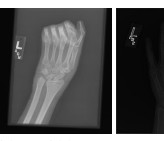 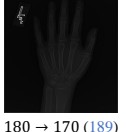

145 → 138 (138)  19 → 11 (12)  28 → 31 (33)    217 → 211 (216) 180 → 170 (189)
(a)                                             (b)

*Figure 5.* (a) Success and (b) failure cases of the label refinement on the RSNA training dataset. Under each image, the noisy, refined, and true ranks are specified: noisy → refined (true).

best competing scores by 1.1 percentage points. Even at the lowest $\kappa = 0.2$, SOL reduces the MAE by 2.6% and improves the CS by 0.3 percentage points over GOL.

**Medical assessment:** In Table 4, we compare the results on the bone age assessment dataset RSNA. The proposed SOL again yields the best results with large margins, with the single exception of the MAE metric for the Laplacian noise. For example, even at $\kappa = 0.1$, SOL outperforms the second-best MWR and GOL with significant gaps of 0.254 and 1.0 in the MAE and CS metrics, respectively. This noise-robustness is meaningful because obtaining error-free annotations on medical datasets is difficult and costly.

**Textual regression with real-world noise:** To further validate the effectiveness of SOL, we apply it to a textual regression task in NLP, where labels are known to be noisy due to subjective human annotations. We use the direct assessment (DA) scores from the Ru-En language pairs in WMT2020 (Specia et al., 2020) as regression targets, and follow Wang et al. (2022a) by adopting the same BERT encoder. As shown in Table 5, SOL achieves the best performance with Pearson's correlation of 0.680 and Spearman's correlation of 0.649, outperforming the previous state-of-the-art by clear margins of 2.0 and 1.9 points, respectively. These results demonstrate that SOL can robustly handle real-world label noise beyond controlled synthetic settings.

### 4.3. Analysis

**Label refinement:** SOL refines noisy ranks present in the training dataset using the outlier detection and relabeling scheme in Section 3.3. Figure 5 shows examples of detected

*Table 5.* Performance comparison on the WMT2020 dataset.

| | REAL-WORLD NOISE | |
| ALGORITHM | PCC($\uparrow$) | SRCC($\uparrow$) |
|---|---|---|
| BASE (WANG ET AL., 2022A) | 0.645 | 0.612 |
| DIS (WANG ET AL., 2022A) | 0.653 | 0.627 |
| RES (WANG ET AL., 2022A) | _0.660_ | _0.630_ |
| SOL | **0.680** | **0.649** |

*Table 6.* Ablation studies for the loss functions in (7) on the CLAP2015 dataset.

| | | | GAUSSIAN | | | | | |
| | | | $\kappa = 0.2$ | | $\kappa = 0.3$ | | $\kappa = 0.4$ | |
| METHOD | $\ell_{disc}$ | $\ell_{order}$ | MAE($\downarrow$) | CS($\uparrow$) | MAE($\downarrow$) | CS($\uparrow$) | MAE($\downarrow$) | CS($\uparrow$) |
|---|---|---|---|---|---|---|---|---|
| I | ✓ | | 20.029 | 14.92 | 16.433 | 20.76 | 18.582 | 17.52 |
| II | | ✓ | 3.586 | 78.41 | 3.785 | 76.74 | 4.044 | 73.40 |
| III | ✓ | ✓ | **3.559** | **78.68** | **3.764** | **77.11** | **4.002** | **73.68** |

outliers in RSNA at $\kappa = 0.15$ (Gaussian). Label errors of up to 10 are well refined in the successful cases in Figure 5(a). In less frequent failure cases, such as Figure 5(b), the refined ranks have bigger errors than the original ones. These are, however, challenging examples because of finger folding or underexposure. More results of the outlier detection and relabeling scheme are provided in Appendices D.4 and D.15.

**Loss functions:** Table 6 compares ablated methods for the loss functions in (7). Method I employs the discriminative loss $\ell_{disc}$ only, while method II does the stochastic order loss $\ell_{order}$ only. Compared with method III (SOL), methods I and II degrade the rank estimation results, indicating that both losses contribute to the performance improvement and are complementary to each other. Note that method I yields poor results, for the discriminative loss alone cannot construct a meaningful embedding space; it is trivial to reduce $\ell_{disc}$ to zero by merging all instances into a single point in the space. However, by comparing II and III, we see that $\ell_{disc}$ helps to sort instances in the embedding space properly by attracting and repelling instances according to their ranks.

## 5. Conclusions

In this work, we addressed the problem of rank estimation under label noise by reframing it as a stochastic ordering problem, in which instance–rank relationships are inherently uncertain rather than deterministic. Based on this perspective, we proposed stochastic order learning (SOL), a learning framework that captures ordinal label uncertainty and learns an embedding space through probabilistic ordering constraints. By combining a discriminative objective for instance–centroid interactions with a stochastic order objective for relational consistency, SOL enables robust rank estimation in the presence of noisy ordinal annotations. Extensive experiments across diverse application domains, including facial age estimation, aesthetic assessment, medical image analysis, and textual regression, demonstrate the effectiveness and generality of the proposed approach under various types and levels of label noise.

## Impact Statement

Due to the intrinsic imbalance of facial datasets (Ricanek & Tesafaye, 2006; Escalera et al., 2015), there may be unwanted gender or racial bias for deep learning-based facial analysis methods. When trained on such facial datasets, the proposed algorithm is not free from this bias either. Thus, the bias should be resolved before any practical usage. We recommend using the proposed algorithm for research only.

## Acknowledgements

This work was supported by the National Research Foundation of Korea (NRF) funded by the Korea Government (MSIT) (No. RS-2024-00397293, RS-2022-NR068986), and by the AI Computing Infrastructure Enhancement (GPU Rental Support) User Support Program funded by MSIT (No. RQT-25-090187).

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

## A. Derivation of Monotonicity Constraint in (5)

The desideratum in (4) can be written as

$$\sum_s p_s d^2(h_x, \mu_{r_x+s}) \le \sum_s p_s d^2(h_x, \mu_{(r_x+k)+s}) \quad \text{for all } k. \tag{23}$$

For simpler notations, let $L_s \triangleq d^2(h_x, \mu_{r_x+s})$. Then, the desideratum is given by

$$\sum_s p_s L_s \le \sum_s p_s L_{s+k} \quad \text{for all } k. \tag{24}$$

First, let us consider the case for $k = 1$. From (24), we have

$$\begin{aligned}
\cdots + p_2 L_{-2} + p_1 L_{-1} + p_0 L_0 + p_1 L_1 + p_2 L_2 + \cdots \quad &\le \\
\cdots + p_3 L_{-2} + p_2 L_{-1} + p_1 L_0 + p_0 L_1 + p_1 L_2 + \cdots &
\end{aligned} \tag{25}$$

since $p_s$ in (2) is symmetric. Thus,

$$(p_0 - p_1)(L_0 - L_1) + (p_1 - p_2)(L_{-1} - L_2) + (p_2 - p_3)(L_{-2} - L_3) + \cdots \le 0. \tag{26}$$

Because $p_s$ in (2) is also unimodal, the coefficients $(p_s - p_{s+1})$ are positive for all $s \ge 0$. Hence, the inequality in (26) is satisfied if

$$L_0 \le L_1, \quad L_{-1} \le L_2, \quad L_{-2} \le L_3, \quad \cdots \tag{27}$$

or equivalently

$$L_{-m} \le L_{1+m} \quad \text{for all } m \ge 0. \tag{28}$$

Next, let us consider the case for $k = 2$. Similar to (26), we have

$$(p_0 - p_2)(L_0 - L_2) + (p_1 - p_3)(L_{-1} - L_3) + (p_2 - p_4)(L_{-2} - L_4) + \cdots \le 0. \tag{29}$$

This is satisfied if

$$L_{1-m} \le L_{1+m} \quad \text{for all } m \ge 0. \tag{30}$$

In general, if $k \ge 1$, we have the following condition:

$$L_{\lfloor \frac{k}{2} \rfloor - m} \le L_{\lceil \frac{k}{2} \rceil + m} \quad \text{for all } m \ge 0. \tag{31}$$

Note that (28) and (30) are special cases of (31). Symmetrically, if $k \le -1$, we have the condition:

$$L_{\lfloor \frac{k}{2} \rfloor - m} \ge L_{\lceil \frac{k}{2} \rceil + m} \quad \text{for all } m \ge 0. \tag{32}$$

Both conditions in (31) and (32) are satisfied if

$$L_0 \le L_{\pm 1} \le L_{\pm 2} \le L_{\pm 3} \le \cdots , \tag{33}$$

implying that $L_k$ should be a monotonic increasing function of $|k|$. Rewriting this monotonicity constraint in the original notations, we have the sufficient condition in (5),

$$d(h_x, \mu_{r_x+s}) \le d(h_x, \mu_{r_x+t}) \quad \text{for all } |s| \le |t|. \tag{34}$$

## B. Derivation of Centroid Rule in (18)

Based on the desideratum in (4), we formulate a cost function

$$\begin{aligned}
J &= \sum_{x \in \mathcal{X}} D_h(x, r_x) \tag{35} \\
&= \sum_{x \in \mathcal{X}} \sum_s p_s d^2(h_x, \mu_{r_x+s}) \tag{36} \\
&= \sum_{x \in \mathcal{X}} \sum_s p_s (\mu_{r_x+s}^T \mu_{r_x+s} - 2h_x^T \mu_{r_x+s} + h_x^T h_x) \tag{37} \\
&= \sum_{x \in \mathcal{X}} \sum_r p_{r-r_x} (\mu_r^T \mu_r - 2h_x^T \mu_r + h_x^T h_x). \tag{38}
\end{aligned}$$

We then update the centroids $\{\mu_r\}_{r=1}^n$ to minimize the cost function $J$. By differentiating $J$ with respect to each $\mu_r$ and setting it to zero, we have

$$\frac{\partial J}{\partial \mu_r} = \sum_{x \in \mathcal{X}} p_{r-r_x}(2\mu_r - 2h_x) = 0. \tag{39}$$

Hence, the optimal centroid is given by

$$\mu_r = \frac{\sum_{x \in \mathcal{X}} p_{r-r_x} h_x}{\sum_{x \in \mathcal{X}} p_{r-r_x}}, \qquad r \in \{1, \ldots, n\}. \tag{40}$$

## C. Implementation Details

### C.1. Datasets

**MORPH II** (Ricanek & Tesafaye, 2006)**:** It is a dataset for facial age estimation, consisting of 55K facial images in the age range $[16, 77]$. It provides age, gender, and race labels. As in Chang et al. (2011), we use 5,492 Caucasian images divided into training and test sets with a ratio of 8:2.

**CLAP2015** (Escalera et al., 2015)**:** It is for apparent age estimation. The apparent age of each image was rated by at least 10 annotators within the range $[3, 85]$, and the mean rating is used as the ground-truth. This dataset provides 4,691 facial images in total that are split into 2,476 for training, 1,136 for validation, and 1,079 for testing.

**AADB** (Kong et al., 2016)**:** It is a dataset for aesthetic score regression, composed of 10,000 photographs of various themes such as scenery and close-up. We use 8,500 images for training, 500 for validation, and 1,000 for testing. Each image is annotated with an aesthetic score in $[0, 1]$. We quantize the continuous scores with a step size of 0.01 to have 101 discrete ranks.

**RSNA** (Halabi et al., 2019)**:** It is for pediatric bone age assessment, containing 14,236 hand radiographs. We employ the official evaluation protocol in Halabi et al. (2019) — 12,611 for training, 1,425 for validation, and 200 for testing. The bone age range is $[0, 216]$ in months.

**WMT2020** (Specia et al., 2020)**:** It is a dataset for machine translation quality estimation, where translations are scored with human direct assessment (DA) on a scale of $[0, 100]$. The dataset includes seven language pairs of varying resource levels, with sentences mostly sourced from Wikipedia. In this work, we use the Russian→English (Ru-En) subset for evaluation.

### C.2. Noise Distribution Settings

To evaluate the robustness of the proposed SOL, we add random noise generated from three different probability distributions: Gaussian, Laplacian, uniform, and skewed. In all cases, the noise magnitude is controlled by adjusting the noise ratio $\kappa$.

1. Gaussian distribution:

$$\mathbf{e} \sim \mathcal{N}(0, (\kappa \cdot \sigma_{\mathcal{X}})^2). \tag{41}$$

2. Laplacian distribution:

$$\mathbf{e} \sim \mathrm{Laplace}(0, \kappa \cdot \sigma_{\mathcal{X}}) \tag{42}$$

with probability density

$$p(e) = \frac{1}{2\kappa \cdot \sigma_{\mathcal{X}}} \exp\left(-\frac{|e|}{\kappa \cdot \sigma_{\mathcal{X}}}\right). \tag{43}$$

3. Uniform distribution:

$$\mathbf{e} \sim \mathcal{U}(-\kappa \cdot \sigma_{\mathcal{X}}, \kappa \cdot \sigma_{\mathcal{X}}). \tag{44}$$

4. Skewed distribution:

$$\mathbf{e} \sim \mathrm{SkewNorm}(a = 5, \mu = 0, \sigma = \kappa \cdot \sigma_{\mathcal{X}}). \tag{45}$$

## C.3. Specification of $\sigma$ in (22)

Table 7 specifies the exact values of $\sigma$ for generating the noise in (22) for each dataset.

*Table 7.* The values of $\sigma$ according to $\kappa$.

| | $\sigma$ | | | | | |
| --- | --- | --- | --- | --- | --- | --- |
| | $\kappa = 0.1$ | $\kappa = 0.15$ | $\kappa = 0.2$ | $\kappa = 0.3$ | $\kappa = 0.4$ | $\kappa = 0.5$ |
| MORPH II | 1.092 | 1.638 | 2.184 | 3.276 | 4.368 | 5.460 |
| CLAP2015 | 1.235 | 1.853 | 2.471 | 3.706 | 4.941 | 6.177 |
| AADB | 0.018 | 0.028 | 0.037 | 0.055 | 0.074 | 0.102 |
| RSNA | 4.118 | 6.177 | 8.326 | 12.355 | 16.473 | 20.591 |

## C.4. Tolerance Values for Computing Cumulative Scores

In facial age estimation, the cumulative score (CS) is commonly measured using a tolerance value of 5 (Chang et al., 2011; Shen et al., 2018). For a fair comparison, we also adopt the tolerance value of 5 for the MORPH II and CLAP2015 datasets.

The ranks in AADB, an aesthetic score regression dataset, range from 0 to 1. Thus, for AADB, we use a tolerance value of 0.25, instead of 5.

In medical assessment, previous work only adopts the MAE metric and does not compute CS scores. Bone ages in the RSNA dataset are measured in months instead of years, so RSNA has a bigger error range than facial age estimation datasets. If the same tolerance value 5 is used, it yields very poor CS scores. Thus, we set the tolerance value to be the smallest integer at which the CS scores exceed 75% for all noise ratios $\kappa$. Based on the results in Table 8, we set 12 as the tolerance value for RSNA in all experiments.

*Table 8.* CS scores (%) of SOL according to the tolerance values on the RSNA dataset (Gaussian label noise).

| Tolerance value | 10 | 11 | **12** | 13 | 14 | 15 | 20 | 25 |
| --- | --- | --- | --- | --- | --- | --- | --- | --- |
| $\kappa = 0.1$ | 71.00 | 75.00 | **78.50** | 82.50 | 84.50 | 87.50 | 94.00 | 97.00 |
| $\kappa = 0.15$ | 68.50 | 74.50 | **80.50** | 85.50 | 86.50 | 89.00 | 95.00 | 97.50 |
| $\kappa = 0.2$ | 69.50 | 73.00 | **76.50** | 80.00 | 84.00 | 86.00 | 92.00 | 99.00 |

We also show the CS curves according to tolerance values on the RSNA dataset in Figure 6. It is observed that the proposed SOL performs better than the state-of-the-art algorithms with the highest area under the curve (AUC) at all noise ratios $\kappa$.

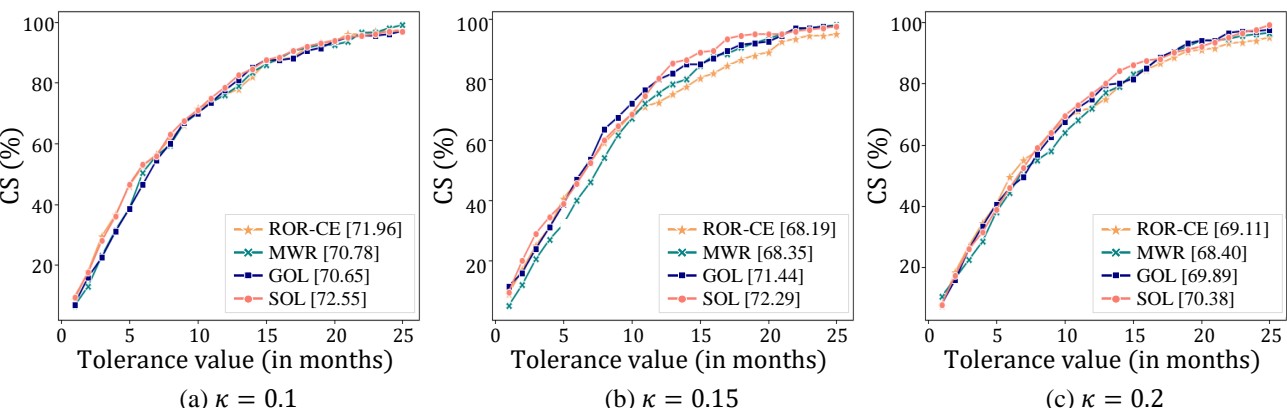

(a) $\kappa = 0.1$        (b) $\kappa = 0.15$        (c) $\kappa = 0.2$

*Figure 6.* Comparison of the CS curves according to tolerance values on the RSNA dataset (Gaussian label noise). The legend of each graph includes the AUC score for the corresponding algorithm.

## C.5. Network Architecture

As described in Section 3.2, we employ an encoder to map each instance into a feature vector in an embedding space. The network structure for the encoder $h$ is specified in Figure 7. The encoder is based on the VGG16 network and takes a $224 \times 224 \times 3$ image as input.

Encoder $h$

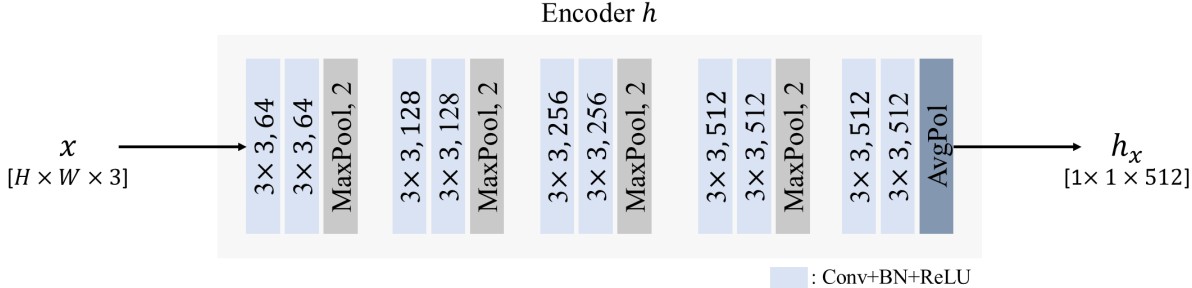

*Figure 7.* Network structure of the encoder $h$.

## C.6. Hyperparameter Settings

For WMT2020, we train the network for 20 epochs. For all the other datasets, we train the network for 100 epochs. Table 9 summarizes the hyperparameters for each dataset.

*Table 9.* Hyperparameter settings

| Dataset | Learning rate | Batch size | $T$ in (8) | $\tau$ in (11) | $\gamma$ in (15) | $\beta$ in (19) | $\sigma_{\text{test}}$ |
|---|---|---|---|---|---|---|---|
| MORPH II | $10^{-4}$ | 32 | 1 | 3 | 0.25 | 0.9 | 1 |
| CLAP2015 | $10^{-4}$ | 32 | 1 | 3 | 0.25 | 0.85 | 1 |
| AADB | $5 \times 10^{-5}$ | 32 | 1 | 5 | 0.25 | 0.85 | 0.01 |
| RSNA | $5 \times 10^{-5}$ | 32 | 1 | 3 | 0.25 | 0.9 | 1 |
| WMT2020 | $2 \times 10^{-5}$ | 16 | 1 | 3 | 0.25 | 0.85 | 1 |

## D. More Experimental Results

In the following experiments, we use Gaussian distributions for label noise.

### D.1. Hyperparameter Analysis

**Analysis on $T$ in** (8): Table 10 compares the MAE scores at different $T$'s on the CLAP2015 dataset. In this test, $\tau = 3$, $\beta = 0.85$, and $\sigma_{\text{test}} = 1$. Except at $\kappa = 0.2$, where the setting $T = 1$ yields a slightly lower MAE by 0.004 than $T = 3$, the best results are provided by the setting $T = 1$. Thus, we set $T = 1$ as the default mode.

*Table 10.* MAE scores according to $T$ on the CLAP2015 dataset.

|              | $T = 1$ | $T = 2$ | $T = 3$ |
|--------------|---------|---------|---------|
| $\kappa = 0.2$ | 3.559 | 3.565 | 3.555 |
| $\kappa = 0.3$ | 3.764 | 3.779 | 3.832 |
| $\kappa = 0.4$ | 4.002 | 4.032 | 4.050 |
| $\kappa = 0.5$ | 4.170 | 4.196 | 4.196 |

**Analysis on $\tau$ in** (11): Table 11 compares the MAE results at different $\tau$'s on CLAP2015. In this test, $T = 1$, $\beta = 0.85$, and $\sigma_{\text{test}} = 1$. Note that $\tau$ is a threshold in (11) to control the balance between rank precision and model robustness. Using $\tau$ as big as 3 achieves robustness and yields decent MAE results. However, when $\tau$ is larger than 3, the performance drops because of the model under-fitting. Hence, we set $\tau = 3$ for CLAP2015.

*Table 11.* MAE scores according to $\tau$ on the CLAP2015 dataset.

|              | $\tau = 1$ | $\tau = 2$ | $\tau = 3$ | $\tau = 4$ |
|--------------|-----------|-----------|-----------|-----------|
| $\kappa = 0.2$ | 3.574 | 3.610 | 3.559 | 3.646 |
| $\kappa = 0.3$ | 3.777 | 3.822 | 3.764 | 3.794 |
| $\kappa = 0.4$ | 4.034 | 3.980 | 4.002 | 4.039 |
| $\kappa = 0.5$ | 4.236 | 4.209 | 4.170 | 4.292 |

**Analysis on $\beta$ in** (19): Table 12 lists the results at different $\beta$'s on CLAP2015. In this test, $T = 1$, $\tau = 3$, and $\sigma_{\text{test}} = 1$. $\beta$ is a parameter to control the precision of outlier detection in (19). Increasing $\beta$ increases the precision, but it also decreases the number of instances that are detected. With a low $\beta$, more instances can be detected as outliers, but there is also the risk of false positives. Generally, the setting $\beta \geq 0.85$ yields better results than $\beta < 0.85$. This is because less precise outlier detection at a low $\beta$ may deteriorate network training by increasing label noise. As specified in Table 9, we set $\beta = 0.85$ for CLAP2015 and AADB and $\beta = 0.9$ for MORPH II and RSNA.

*Table 12.* MAE scores according to $\beta$ on CLAP2015.

|              | $\beta = 0.8$ | $\beta = 0.85$ | $\beta = 0.9$ | $\beta = 0.95$ |
|--------------|--------------|---------------|--------------|---------------|
| $\kappa = 0.2$ | 3.566 | 3.559 | 3.544 | 3.570 |
| $\kappa = 0.3$ | 3.849 | 3.764 | 3.797 | 3.804 |
| $\kappa = 0.4$ | 4.070 | 4.002 | 4.036 | 4.062 |
| $\kappa = 0.5$ | 4.173 | 4.170 | 4.177 | 4.171 |

**D.2. Analysis on $\sigma_{\text{test}}$**

**Gaussian noise assumption and fixed $\sigma_{\text{test}}$:** Many real-world rank-estimation datasets, including CLAP2015 (Escalera et al., 2015), AADB (Kong et al., 2016), and RSNA (Halabi et al., 2019), obtain their ground-truth labels by averaging multiple independent human annotations. Due to the central-limit effect, such averaged labels empirically follow a Gaussian-like distribution; CLAP2015 further provides per-sample variance estimates that directly support this assumption. While individual annotators may deviate from Gaussian behavior, the aggregated labels are typically well approximated by a Gaussian model, making the discrete Gaussian noise formulation in (2) a reasonable choice.

In practice, the true standard deviation of annotation noise is unknown at test time. Therefore, SOL uses a fixed $\sigma_{\text{test}}$ to compute the probabilities $p_s$ in (2). The following analysis evaluates how sensitive SOL is to this hyperparameter.

**Sensitivity to $\sigma_{\text{test}}$:** We examine how the performance of SOL changes with different choices of the fixed $\sigma_{\text{test}}$ used to compute $p_s$ in (2). Table 13 summarizes the MAE results on the CLAP2015 dataset under $T = 1$, $\tau = 3$, and $\beta = 0.85$. A larger $\sigma_{\text{test}}$ couples each instance $x$ more strongly with distant rank centroids, which can weaken rank discrimination. In contrast, a very small value makes the model sensitive to label errors because $x$ interacts only with nearby centroids. Balancing these effects, $\sigma_{\text{test}} = 1.0$ provides the most stable performance in most settings.

*Table 13.* MAE results according to $\sigma_{\text{test}}$ on the CLAP2015 dataset .

|  | $\sigma_{\text{test}} = 0.5$ | $\sigma_{\text{test}} = 1.0$ | $\sigma_{\text{test}} = 1.5$ | $\sigma_{\text{test}} = 2.0$ | $\sigma_{\text{test}} = 2.5$ | $\sigma_{\text{test}} = 3.0$ | $\sigma_{\text{test}} = 3.5$ |
|---|---|---|---|---|---|---|---|
| $\kappa = 0.2$ | 3.555 | 3.559 | 3.548 | 3.549 | 3.588 | 3.593 | 3.670 |
| $\kappa = 0.3$ | 3.801 | 3.764 | 3.794 | 3.797 | 3.848 | 3.888 | 3.985 |
| $\kappa = 0.4$ | 4.000 | 4.002 | 4.072 | 4.070 | 4.061 | 4.194 | 4.355 |
| $\kappa = 0.5$ | 4.198 | 4.170 | 4.203 | 4.288 | 4.259 | 4.343 | 4.499 |

We plot the MAE scores according to $\sigma_{\text{test}}$ in Figure 8. It is observed that MAE results start to degrade significantly once $\sigma_{\text{test}} \geq 4.0$. As shown in Figure 9, the probability distribution $p_s$ in (2) flattens as $\sigma_{\text{test}}$ gets bigger. Thus, the probabilities assigned to different ranks become indistinguishable for SOL to operate well when $\sigma_{\text{test}} \geq 4.0$. Hence, it is appropriate to use a $\sigma_{\text{test}}$ less than 4.0 for CLAP2015.

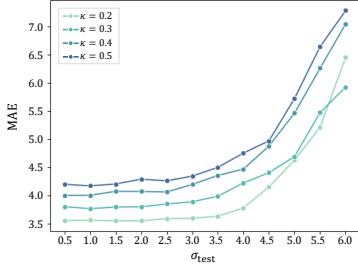

*Figure 8.* MAE according to $\sigma_{\text{test}}$ on CLAP2015.

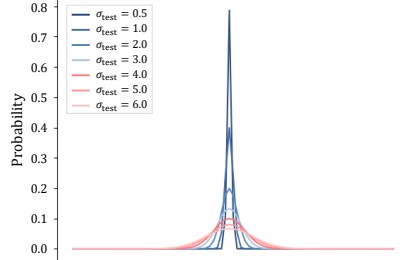

*Figure 9.* $p_s$ in (2) for different $\sigma_{\text{test}}$.

Table 14 shows a similar trend on the WMT2020 dataset. Although the evaluation metrics differ (PCC and SRCC), the overall variation with respect to $\sigma_{\text{test}}$ remains small, confirming that SOL is not highly sensitive to this hyperparameter in real-world settings. Finally, the $\sigma_{\text{test}}$ values used for all datasets in the main paper are summarized in Table 9.

*Table 14.* PCC and SRCC scores of SOL on the WMT2020 dataset for different values of $\sigma_{\text{test}}$.

| $\sigma_{test}$ | 0.5 | 1.0 | 1.5 | 2.0 | 2.5 | 3.0 | 3.5 | 4.0 |
|---|---|---|---|---|---|---|---|---|
| PCC (↑) | 0.664 | 0.680 | 0.672 | 0.679 | 0.672 | 0.670 | 0.675 | 0.683 |
| SRCC (↑) | 0.639 | 0.649 | 0.640 | 0.654 | 0.656 | 0.641 | 0.646 | 0.653 |

**Adaptive $\sigma_{\text{test}}$:** To examine whether $\sigma$ can be estimated from data, we add a lightweight head that predicts the mean $\mu$ and standard deviation $\sigma$, trained with a Gaussian negative log-likelihood loss, so that the predicted $\sigma$ replaces the constant in (2). We evaluate two variants: *Joint training*, where the $\sigma$-prediction head and SOL are optimized together, and *Two-stage scheme*, where the $\sigma$-prediction head is trained first and then frozen during SOL training. As shown below for CLAP2015 at $\kappa = 0.4$, the fixed setting achieves better MAE and CS than both adaptive variants.

*Table 15.* Comparison of adaptive $\sigma_{\text{test}}$ strategies on the CLAP2015 dataset at $\kappa = 0.4$.

| Method | MAE ($\downarrow$) | CS ($\uparrow$) |
|---|---|---|
| Joint adaptive $\sigma_{\text{test}}$ | 5.032 | 67.10 |
| Two-stage adaptive $\sigma_{\text{test}}$ | 4.171 | 71.64 |
| Fixed $\sigma_{\text{test}}$ (default) | **4.002** | **73.68** |

**Input-dependent noise:** To assess the robustness of SOL beyond global perturbation models, we additionally evaluate an input-dependent noise setting motivated by a common observation in age estimation: labels for very young or elderly subjects are relatively easier to estimate, whereas middle-aged samples are often more ambiguous and thus more prone to annotation noise. Specifically, we define a rank-dependent modulation $f_{\text{rank}}(r) = \exp\left(-\frac{(r-r_{\text{peak}})^2}{2w^2}\right)$, with $r_{\text{peak}} = 40$ and $w = 15$, and set the instance-level noise as $\sigma(x) = \kappa \cdot \sigma_{\mathcal{X}} \cdot (1 + f_{\text{rank}}(r_x))$, which yields higher noise around ambiguous age ranges. As shown in Table 16, SOL continues to outperform the strong GOL baseline on both MAE and CS. These results indicate that SOL remains robust under input-dependent noise, beyond a single global noise model.

*Table 16.* Performance comparison on MORPHII dataset at $\kappa = 0.3$ under input-dependent noise.

| Method | MAE ($\downarrow$) | CS ($\uparrow$) |
|---|---|---|
| GOL | 2.724 | 88.25 |
| SOL | **2.702** | **88.71** |

### D.3. Loss Functions

**Alternatives to $\ell_{\text{disc}}$ in (8):** Table 17 compares alternative loss terms for $\ell_{\text{disc}}$. Method I, which is also known as the center loss, aims at directly locating an instance $x$ close to its corresponding centroid $\mu_{r_x}$. On the other hand, method II decreases not only the distance to the corresponding centroid but also to its stochastically-related centroids. Method II performs better than method I. However, the table shows that the proposed discriminative loss $\ell_{\text{disc}}$ yields the best performance.

*Table 17.* Comparison of alternative choices for $\ell_{\text{disc}}$ in (8) on the CLAP2015 dataset at $\kappa = 0.2$.

| Method | Alternative to $\ell_{\text{disc}}$ | MAE ($\downarrow$) |
|---|---|---|
| I | $d(h_x, \mu_{r_x})$ | 3.593 |
| II | $D_h(x, r_x)$ | 3.585 |
| III | $\ell_{\text{disc}}$ in (8) | 3.559 |

### D.4. Outlier Detection and Relabeling

**Impacts of label refinement:** To show the effectiveness of the proposed label refinement (*i.e.* outlier detection and relabeling) scheme, Table 18 compares the results of SOL with and without the label refinement, respectively, on CLAP2015. By examining Table 18 together with Table 2, it can be observed that even without the refinement SOL outperforms the conventional algorithms. However, by applying the refinement scheme, the proposed SOL further improves overall performance. In general, the label refinement reduces label noise in a training dataset, making the training process more reliable. The impact of relabeling also depends on dataset size. Because CLAP2015 is relatively small, only a few samples are identified as outliers, so the quantitative improvements are modest. In contrast, larger datasets such as RSNA contain more inconsistent labels, making the refinement more beneficial. The RSNA results in Table 19 clearly demonstrate this tendency.

*Table 18.* Comparison of the proposed SOL with and without the label refinement on CLAP2015.

|  | $\kappa = 0.2$ | | $\kappa = 0.3$ | | $\kappa = 0.4$ | | $\kappa = 0.5$ | |
| --- | --- | --- | --- | --- | --- | --- | --- | --- |
| Algorithm | MAE ($\downarrow$) | CS ($\uparrow$) | MAE ($\downarrow$) | CS ($\uparrow$) | MAE ($\downarrow$) | CS ($\uparrow$) | MAE ($\downarrow$) | CS ($\uparrow$) |
| w/o label refinement | **3.556** | 78.41 | 3.766 | 76.37 | 4.058 | **73.68** | 4.208 | **72.57** |
| w/ label refinement | 3.559 | **78.68** | **3.764** | **77.11** | **4.002** | **73.68** | **4.170** | 71.64 |

*Table 19.* Comparison of the proposed SOL with and without the label refinement on RSNA.

|  | $\kappa = 0.10$ | | $\kappa = 0.15$ | | $\kappa = 0.20$ | |
| --- | --- | --- | --- | --- | --- | --- |
| Algorithm | MAE ($\downarrow$) | CS ($\uparrow$) | MAE ($\downarrow$) | CS ($\uparrow$) | MAE ($\downarrow$) | CS ($\uparrow$) |
| w/o label refinement | 7.967 | **81.50** | 7.800 | 79.50 | 8.196 | 74.00 |
| w/ label refinement | **7.579** | 78.50 | **7.706** | **80.50** | **8.051** | **76.50** |

**Alternative relabeling schemes:** In the proposed relabeling scheme, the ranks of detected outliers are adjusted by the same magnitude via (20). Here, we assess the performance when each detected outlier is relabeled using different magnitudes. Specifically, we adjust the rank of each outlier instance by half of the absolute difference between its noisy and estimated rank. Table 20 lists the results on the CLAP2015 dataset. Compared to method I performing no relabeling, method II improves MAE. However, the proposed relabeling scheme provides the best results. Using the same average value to adjust the ranks prevents drastic changes in rank labels, yielding more reliable performance.

*Table 20.* Analysis on the relabeling scheme on the CLAP2015 dataset at $\kappa = 0.4$.

|  | Relabeling schemes | MAE ($\downarrow$) | CS ($\uparrow$) |
| --- | --- | --- | --- |
| I | No relabeling | 4.058 | **73.68** |
| II | Different magnitudes | 4.012 | 72.75 |
| III | Proposed | **4.002** | **73.68** |

**Noise reduction:** The proposed SOL can refine noisy ranks. To demonstrate this capability, we report MAEs between a noisy rank $r_x$ and the true rank $\bar{r}_x$ and the standard deviations of such noise levels before and after the label refinement in Table 21. In this test, we use the MORPH II and CLAP2015 datasets. Note that the MAE or the standard deviation is reduced in 11 out of 12 tests, confirming the effectiveness of the label refinement. For further analysis, we test how the refinement changes the number of instances at each noise level (*i.e.* label error). Figure 10 plots such statistics on MORPH II at various $\kappa$'s. The red boxes in Figure 10 specify the numbers of instances with high noise levels. We see that the numbers of instances with extreme noise levels are reduced in general. Especially, at $\kappa = 0.4$, the number of instances with $2 \leq e_x \leq 4$ is increased, while that with $e_x \geq 7$ is reduced significantly. It is desirable because severe label errors hinder the construction of a well-sorted embedding space. Consequently, the label refinement generally boosts the performance of SOL.

*Table 21.* Comparison of the average noise levels before and after the label refinement.

| | MORPH II | | | | CLAP2015 | | | |
|---|---|---|---|---|---|---|---|---|
| Noise ratio | MAE | | Standard Deviation | | MAE | | Standard Deviation | |
| $\kappa = 0.2$ | 1.737 | $\rightarrow$ 1.718 | 1.361 | $\rightarrow$ 1.343 | 1.961 | $\rightarrow$ 1.959 | 1.508 | $\rightarrow$ 1.537 |
| $\kappa = 0.3$ | 2.599 | $\rightarrow$ 2.534 | 1.991 | $\rightarrow$ 1.942 | 2.970 | $\rightarrow$ 2.896 | 2.262 | $\rightarrow$ 2.254 |
| $\kappa = 0.4$ | 3.504 | $\rightarrow$ 3.401 | 2.638 | $\rightarrow$ 2.499 | 4.006 | $\rightarrow$ 3.793 | 3.038 | $\rightarrow$ 2.899 |

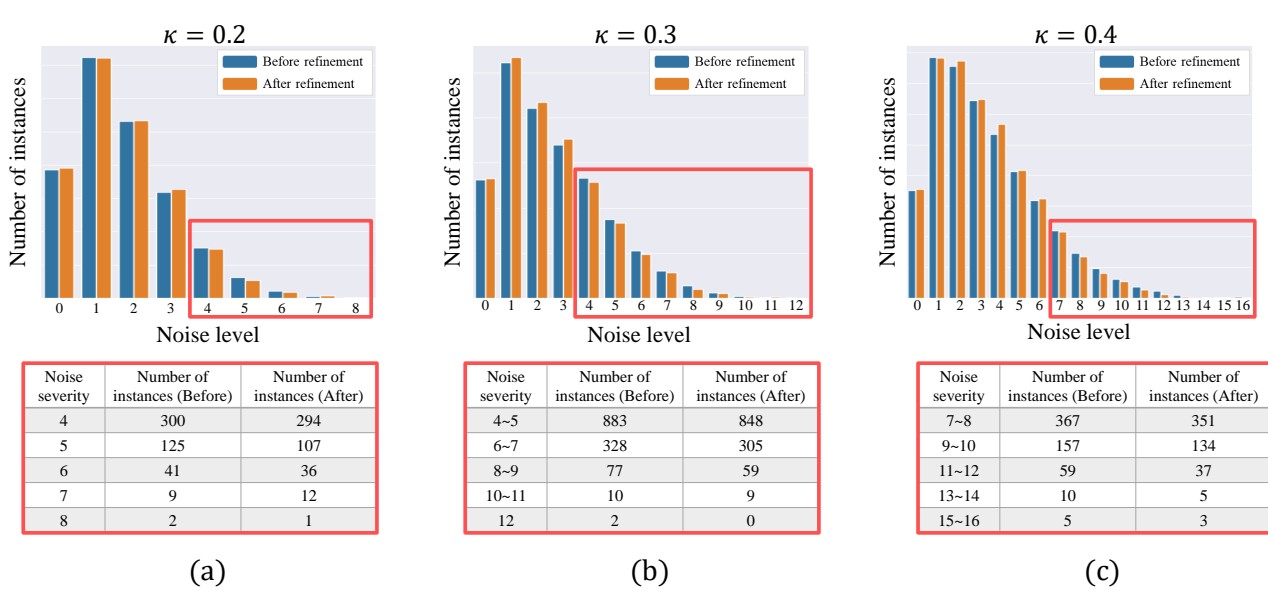

*Figure 10.* Comparison of the numbers of instances at each noise level before and after the label refinement on the MORPH II dataset.

## D.5. Outliers in the WMT2020 Dataset

We provide a qualitative analysis of outlier cases detected by SOL on the real-noise WMT2020 translation-quality dataset. Unlike synthetic noise, discrepancies in WMT2020 originate from genuine human variability, including strong penalties applied to fluent translations and unexpectedly high scores assigned to mistranslated or semantically incorrect outputs. Typical outliers are categorized into two classes.

- Type A: fluent or semantically acceptable translations that receive abnormally low human scores,
- Type B: mistranslated or semantically incorrect outputs that nevertheless receive unusually high scores.

Table 22 presents representative examples identified by SOL. Each case exhibits a clear mismatch between linguistic quality and the annotated score, highlighting the presence of nontrivial and asymmetric annotation noise in WMT2020.

*Table 22.* Representative outliers detected by SOL on the WMT2020 dataset.

| Type | Real Score | Pred Score | Source Text | Translation | Issue |
|---|---|---|---|---|---|
| A1 | 4 | 22 | Ne po cheloveku spes'. | Don't rush into it. | Fluent sentence but unusually low human score. |
| A2 | 6 | 17 | Ne penyay na zerkalo, kol' rozha kriva. | Don't foam at the mirror if it's crooked. | Acceptable fluency, score is unrealistically low. |
| B1 | 66 | 6 | Zadkom, kuvyrkom, da i pod gorku. | Backward, somersault, and downhill. | Literal mistranslation; idiomatic meaning ("things going downhill") is lost. |
| B2 | 56 | 8 | Religiya yad – beregi rebyat. | Religion Poison – Save the Children | Ungrammatical; missing verb ("Religion is poison"), resulting in awkward phrasing. |
| B3 | 67 | 15 | Chto za chudak, da i chudilo. | What a freak, and a miracle. | Semantic error; "chudilo" mistranslated as "miracle," losing intended meaning. |

### D.6. Performance on Partially Noisy Data

In real-world settings, information on which samples are noisy is not given. Hence, for practical use, we assume that all samples have the risk of labeling errors in the experiments in the main paper. However, the proposed SOL is also effective when only a subset of samples are mislabeled. In Table 23, we randomly sample $\varepsilon\%$ of the total dataset and add noise to their labels. The rest of the data is left clean. We compare the proposed SOL to the state-of-the-art algorithm GOL (Lee et al., 2022). In this partially noisy case as well, the proposed SOL generally achieves better performance than GOL.

*Table 23.* MAE results of GOL / SOL on CLAP2015 when only parts of the total data are corrupted.

|  | $\kappa = 0.2$ | $\kappa = 0.3$ | $\kappa = 0.4$ | $\kappa = 0.5$ |
|---|---|---|---|---|
| $\varepsilon = 10$ | 3.442 / **3.420** | 3.540 / **3.505** | 3.590 / **3.549** | 3.690 / **3.639** |
| $\varepsilon = 20$ | 3.492 / **3.471** | 3.568 / **3.547** | 3.561 / **3.536** | 3.605 / **3.572** |
| $\varepsilon = 30$ | 3.498/ **3.480** | 3.591 / **3.588** | **3.612** / 3.631 | 3.731 / **3.696** |
| $\varepsilon = 40$ | **3.510** / 3.518 | 3.657 / **3.607** | 3.736 / **3.731** | **3.737** / 3.762 |
| $\varepsilon = 50$ | 3.497 / **3.495** | 3.715/ **3.704** | 3.784 / **3.710** | 3.778 / **3.737** |

### D.7. Complexity

**Training time:** Table 24 reports the training time per epoch on the CLAP2015 dataset using an RTX 4090 GPU. We also report the additional runtime introduced by SOL due to its stochastic distance computation and label refinement, by employing GOL as the non-stochastic baseline. While SOL introduces an additional computational cost, it remains practical for training.

*Table 24.* Training time per epoch on CLAP2015.

| Algorithm | Training time (s) |
|---|---|
| RankNet | 44.8 |
| SoftRank | 96.2 |
| MWR | 77.3 |
| GOL (non-stochastic) | 27.8 |
| SOL w/o refinement | 39.2 |
| SOL | 52.1 |

We also compare GPU memory usage for loss computation (batch size = 32) in Table 25. GOL consumes substantially more memory, for it constructs full pairwise direction tensors and expanded index structures, which create large intermediate buffers. In contrast, SOL computes pairwise probabilities on the fly without forming dense tensors, resulting in a much smaller memory footprint.

*Table 25.* GPU memory consumption for loss computation (batch size = 32).

| Algorithm | Memory |
|---|---|
| GOL | 8.19 MB |
| SOL | 0.60 MB |

Table 26 compares the times for computing the centroids in (18) to the total training times. Even for the RSNA dataset consisting of 12,611 training samples, it takes only a few minutes to compute the centroids. This is fast enough for most use cases since the centroids are updated only once per epoch.

*Table 26.* The processing times (s) required for training one epoch.

|  | MORPH II | CLAP2015 | AADB | RSNA |
|---|---|---|---|---|
| Centroid computation | 6.1 | 5.1 | 39.2 | 286.1 |
| Training 1 epoch | 60.2 | 52.1 | 145.4 | 1160.7 |

**Training speed-up:** Although the centroid computation is not a major bottleneck, its cost can be further reduced by sub-sampling the training instances used during centroid updates. Table 27 reports the MAE performance and the corresponding time complexities for different sampling ratios.

*Table 27.* Sub-sampling for centroid computation on the CLAP2015 dataset at $\kappa = 0.4$.

| Sampling ratio | MAE | Centroid computation time (s) | Training time per epoch (s) |
|:---:|:---:|:---:|:---:|
| 0.1 | 4.029 | 0.9 | 47.9 |
| 0.2 | 4.018 | 1.2 | 48.2 |
| 1.0 | 4.002 | 5.1 | 52.1 |

Computing the stochastic distances in FP16 further reduces runtime with negligible impact on MAE, as shown in Table 28.

*Table 28.* Mixed-precision computation on the CLAP2015 dataset at $\kappa = 0.4$.

| Precision | MAE | Training time per epoch (s) |
|:---:|:---:|:---:|
| FP16 | 4.008 | 48.0 |
| FP32 | 4.002 | 52.1 |

**Training time on RSNA:** Table 29 compares the per-epoch training costs on the RSNA dataset.

*Table 29.* Training time per epoch on the RSNA dataset.

| Algorithm | Training time per epoch (s) |
|:---:|:---:|
| MWR | 1036.3 |
| GOL | 664.1 |
| SOL | 1160.7 |

The large per-epoch cost of SOL on RSNA is due to the data-loading configuration rather than the loss itself. For comparability with prior studies, all methods were evaluated with `num_workers = 1`, which introduces an I/O bottleneck. As shown in Table 30, enabling standard parallel data loading reduces the time from 1160.7s to 223.6s. The previously reported 1160.7s therefore represents a conservative upper bound caused by serial loading; SOL trains efficiently under typical parallel pipelines.

*Table 30.* Effect of data-loading parallelization on SOL training time for the RSNA dataset.

| num_workers | Training time per epoch (s) |
|:---:|:---:|
| 1 | 1160.7 |
| 8 | 223.6 |

**Testing time:** We also compare the average processing time required for testing a single image in Table 31. We use an RTX 4090 GPU and test on the CLAP2015 dataset. For efficiency, we extract the features of all training images and compute the centroids in advance. Therefore, during the test, only the feature extraction of a test image is required. Note that GOL uses $k$-NN while SOL uses the nearest expectation as the inference rule. Compared to GOL, SOL achieves faster inference.

*Table 31.* The processing times (s) required for testing a single image on CLAP2015.

| Algorithm | Feature extraction (s) | Inference (s) | Total (s) |
|:---:|:---:|:---:|:---:|
| GOL | 0.040 | 0.083 | 0.123 |
| SOL | 0.040 | 0.051 | 0.091 |

**Memory efficiency:** For large-scale training, memory efficiency is also important. Hence, we compare the number of parameters of SOL with those of conventional methods in Table 32. SOL requires the fewest parameters, indicating its potential for large-scale applications.

*Table 32.* Comparison of the network complexity.

| Algorithm | # of parameters |
|---|---|
| ACL (Ye et al., 2023) | 134.68M |
| MWR (Shin et al., 2022) | 139.41M |
| GOL (Lee et al., 2022) | 14.75M |
| SOL | 14.72M |

### D.8. Influence of Label Noise at Different Noise Ratios $\kappa$

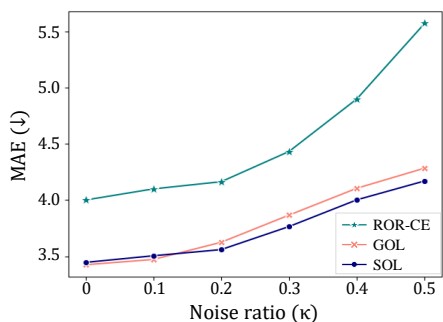

*Figure 11.* MAE results according to the noise ratio $\kappa$ on CLAP2015.

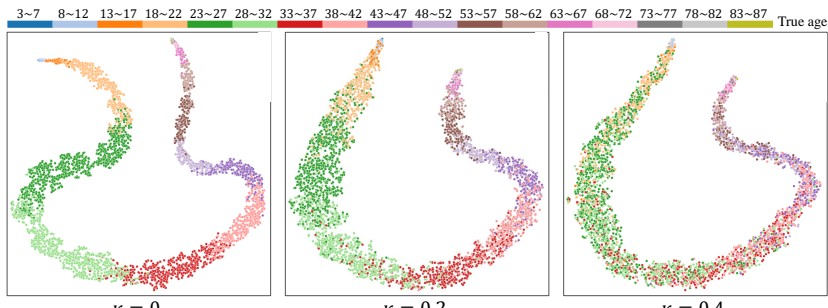

*Figure 12.* t-SNE visualization of the embedding spaces for the CLAP2015 dataset at different noise ratios $\kappa$.

**Noise ratios:** Figure 11 analyzes the influence of label noise on the CLAP2015 dataset, by comparing the proposed SOL with ROR-CE and GOL at different noise ratios $\kappa$. For each algorithm, the increase in $\kappa$ degrades the MAE performance. However, the degradation of the conventional algorithms is severer than that of SOL, demonstrating the superior noise-robustness of SOL.

**Embedding spaces:** Figure 12 visualizes the embedding spaces of SOL using t-SNE (Maaten & Hinton, 2008). As $\kappa$ increases, different ages are more mixed up in the space due to bigger label errors. However, at all $\kappa$, the instances are generally well aligned according to their true ages. We show more t-SNE visualizations in Appendix D.13.

### D.9. Comparison to Learning-to-Rank Methods

For a more complete comparison with learning-to-rank techniques, we additionally implemented RankNet (Burges et al., 2005) and SoftRank (Taylor et al., 2008) under our experimental setup. Both models were trained using the same VGG16 backbone and evaluated through k-NN regression. The results on the MORPH II dataset are summarized in Table 33.

*Table 33.* Comparison with RankNet and SoftRank on the MORPH II dataset.

| | Gaussian | | | | | | Laplacian | | Uniform | | Skewed | |
|---|---|---|---|---|---|---|---|---|---|---|---|---|
| | $\kappa = 0.2$ | | $\kappa = 0.3$ | | $\kappa = 0.4$ | | $\kappa = 0.3$ | | $\kappa = 0.3$ | | $\kappa = 0.3$ | |
| Algorithm | MAE($\downarrow$) | CS($\uparrow$) | MAE($\downarrow$) | CS($\uparrow$) | MAE($\downarrow$) | CS($\uparrow$) | MAE($\downarrow$) | CS($\uparrow$) | MAE($\downarrow$) | CS($\uparrow$) | MAE($\downarrow$) | CS($\uparrow$) |
| RankNet (Burges et al., 2005) | 2.639 | 89.80 | 2.990 | 86.16 | 3.116 | 82.79 | 3.146 | 84.15 | 2.634 | 88.89 | 3.490 | 80.97 |
| SoftRank (Taylor et al., 2008) | 3.147 | 83.06 | 3.394 | 81.97 | 3.427 | 80.15 | 3.801 | 75.96 | 3.137 | 84.34 | 4.018 | 73.32 |
| SOL | **2.489** | **91.35** | **2.663** | **89.62** | **2.826** | **87.70** | **2.794** | **86.89** | **2.499** | **90.89** | **3.296** | **83.15** |

## D.10. Comparison to Label Distribution Learning Methods

SOL and Label Distribution Learning(LDL) differ in formulation, Gaussian usage, and learning objective. LDL focuses on matching each instance to a label distribution, whereas SOL seeks the ordinal structure most consistent with data under corrupted labels. Although both may use Gaussian functions, SOL uses the Gaussian to model corruption among neighboring ranks, not to define per-instance supervision targets. Thus, SOL goes beyond target matching by learning ordering-aware representations and ordinal geometry under noisy ordinal labels.

Table 34 compares SOL with representative LDL methods, including DLDL-v2 (Gao et al., 2018), Uni-Con (Li et al., 2022a), and DHRL (Suzuki et al., 2026), under multiple noise settings on the MORPH II dataset. SOL consistently outperforms all LDL baselines across noise types and levels. This supports our claim that the difference is not merely terminological, but lies in how uncertainty is modeled. LDL captures ambiguity around the observed label, whereas SOL explicitly models label corruption over neighboring ranks. The advantage is especially clear at higher noise levels (*e.g.* $\kappa = 0.4$).

*Table 34.* Comparison with label distribution learning methods on the MORPH II dataset.

| | Gaussian | | | | | | Laplacian | | Uniform | | Skewed | |
| | $\kappa = 0.2$ | | $\kappa = 0.3$ | | $\kappa = 0.4$ | | $\kappa = 0.3$ | | $\kappa = 0.3$ | | $\kappa = 0.3$ | |
| Algorithm | MAE($\downarrow$) | CS($\uparrow$) | MAE($\downarrow$) | CS($\uparrow$) | MAE($\downarrow$) | CS($\uparrow$) | MAE($\downarrow$) | CS($\uparrow$) | MAE($\downarrow$) | CS($\uparrow$) | MAE($\downarrow$) | CS($\uparrow$) |
|---|---|---|---|---|---|---|---|---|---|---|---|---|
| DLDL-v2 (Gao et al., 2018) | 2.753 | 84.34 | 2.882 | 84.79 | 3.206 | 77.60 | 3.199 | 78.42 | 2.789 | 83.79 | 3.364 | 77.69 |
| Uni-Con (Li et al., 2022a) | 2.737 | 85.43 | 2.835 | 85.06 | 3.213 | 78.87 | 3.247 | 78.05 | 2.714 | 85.52 | 3.303 | 79.51 |
| DHRL (Suzuki et al., 2026) | 2.609 | 70.49 | 2.737 | 85.34 | 2.870 | 82.60 | 2.992 | 80.41 | 2.617 | 86.70 | 3.305 | 78.05 |
| SOL | **2.489** | **91.35** | **2.663** | **89.62** | **2.826** | **87.70** | **2.794** | **86.89** | **2.499** | **90.89** | **3.296** | **83.15** |

## D.11. Ablation Studies and Analysis on Additional Datasets

To verify that the same design choices transfer beyond CLAP2015, we conducted ablation studies on RSNA (Gaussian noise with $\kappa = 0.15$) and WMT2020. As summarized in Table 35, both datasets follow the same pattern observed earlier: using either $l_{disc}$ or $l_{order}$ alone provides partial performance gains, whereas combining both terms yields the best results.

*Table 35.* Ablation studies on RSNA and WMT2020.

| | | | RSNA | | WMT2020 | |
| Method | $l_{disc}$ | $l_{order}$ | MAE ($\downarrow$) | CS ($\uparrow$) | PCC ($\uparrow$) | SRCC ($\uparrow$) |
|---|---|---|---|---|---|---|
| I | ✓ | | 8.357 | 74.50 | 0.396 | 0.354 |
| II | | ✓ | 8.040 | 77.50 | 0.673 | 0.634 |
| III | ✓ | ✓ | **7.706** | **80.50** | **0.680** | **0.649** |

## D.12. Multi-seed Stability Analysis

To assess the stability of the proposed method, we additionally conduct multi-seed experiments on the MORPH II dataset. Table 36 reports the mean and standard deviation over five random seeds across all noise settings. SOL consistently achieves the best performance in terms of both MAE and CS, while maintaining low variance across random seeds. These results indicate that the performance gains of SOL are stable and are not attributable to favorable random initialization.

*Table 36.* Multi-seed results on the MORPH II dataset. We report the mean and standard deviation over five random seeds.

| | Gaussian | | | | | | Laplacian | | Uniform | | Skewed | |
| | $\kappa = 0.2$ | | $\kappa = 0.3$ | | $\kappa = 0.4$ | | $\kappa = 0.3$ | | $\kappa = 0.3$ | | $\kappa = 0.3$ | |
| Algorithm | MAE($\downarrow$) | CS($\uparrow$) | MAE($\downarrow$) | CS($\uparrow$) | MAE($\downarrow$) | CS($\uparrow$) | MAE($\downarrow$) | CS($\uparrow$) | MAE($\downarrow$) | CS($\uparrow$) | MAE($\downarrow$) | CS($\uparrow$) |
|---|---|---|---|---|---|---|---|---|---|---|---|---|
| MWR (Shin et al., 2022) | $2.574 \pm 0.019$ | $89.816 \pm 0.505$ | $2.708 \pm 0.014$ | $88.100 \pm 0.355$ | $2.857 \pm 0.030$ | $87.012 \pm 0.720$ | $2.885 \pm 0.033$ | $86.760 \pm 0.233$ | $2.529 \pm 0.010$ | $90.674 \pm 0.257$ | $3.376 \pm 0.053$ | $80.326 \pm 0.565$ |
| GOL (Lee et al., 2022) | $2.520 \pm 0.023$ | $90.802 \pm 0.440$ | $2.707 \pm 0.027$ | $88.760 \pm 0.509$ | $2.855 \pm 0.016$ | $86.468 \pm 0.564$ | $2.853 \pm 0.012$ | $86.614 \pm 0.416$ | $2.520 \pm 0.007$ | $90.346 \pm 0.669$ | $3.357 \pm 0.026$ | $82.514 \pm 0.675$ |
| SOL | $\mathbf{2.497 \pm 0.006}$ | $\mathbf{90.984 \pm 0.456}$ | $\mathbf{2.666 \pm 0.016}$ | $\mathbf{89.200 \pm 0.547}$ | $\mathbf{2.831 \pm 0.013}$ | $\mathbf{87.376 \pm 0.230}$ | $\mathbf{2.809 \pm 0.011}$ | $\mathbf{87.504 \pm 0.522}$ | $\mathbf{2.497 \pm 0.008}$ | $\mathbf{91.057 \pm 0.427}$ | $\mathbf{3.317 \pm 0.012}$ | $\mathbf{83.096 \pm 0.589}$ |

## D.13. More t-SNE Visualizations

We visualize the embedding spaces according to different noise ratios $\kappa$ using t-SNE. The t-SNE plots for the MORPH II, AADB, and RSNA datasets are shown in Figures 13, 14, and 15, respectively.

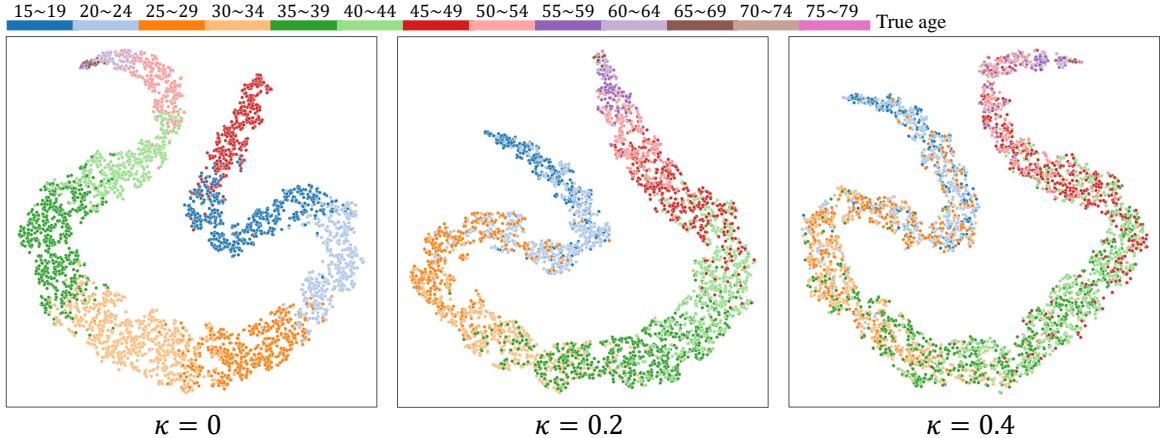

*Figure 13.* t-SNE visualization of the embedding spaces for MORPH II at different noise ratios $\kappa$.

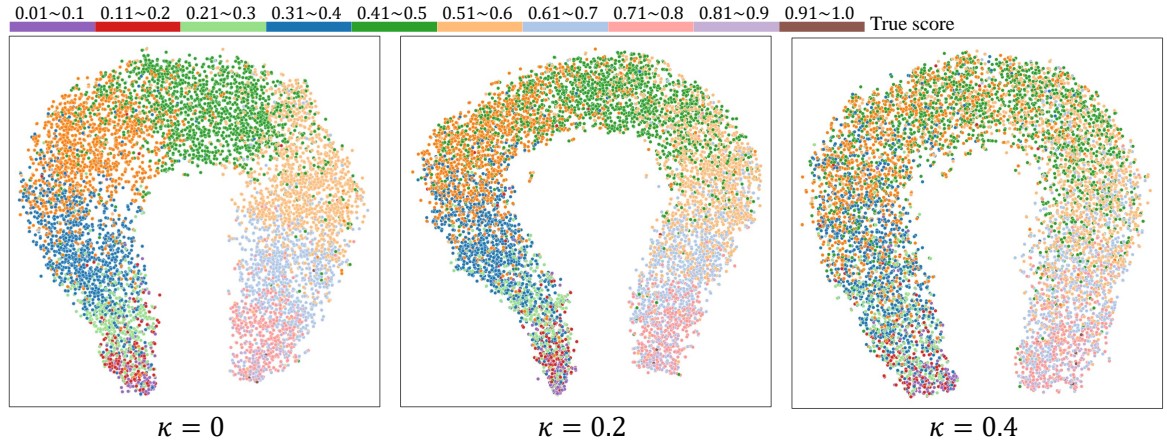

*Figure 14.* t-SNE visualization of the embedding spaces for AADB at different noise ratios $\kappa$.

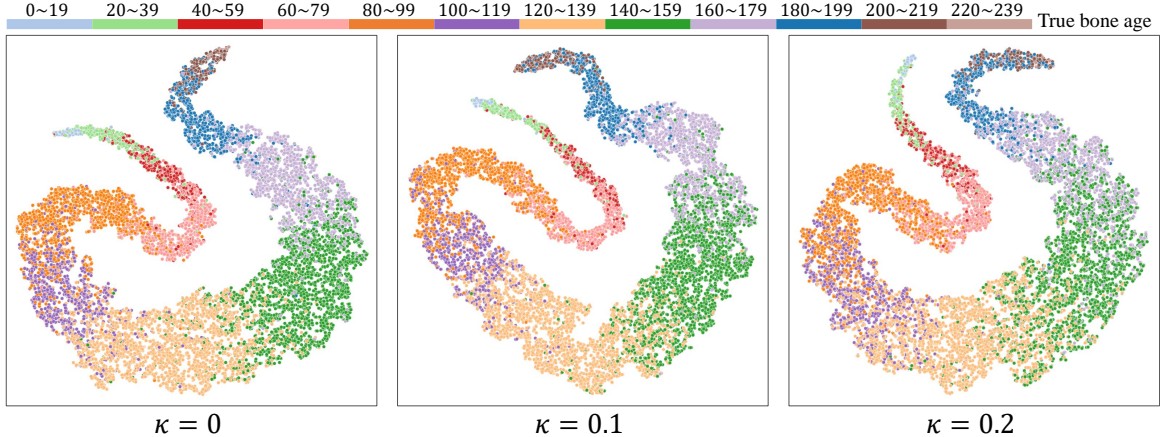

*Figure 15.* t-SNE visualization of the embedding spaces for RSNA at different noise ratios $\kappa$.

### D.14. More Rank Estimation Examples

Figures 16, 17, and 18 show rank estimation results of the proposed SOL on the CLAP2015, AADB, and RSNA datasets, respectively.

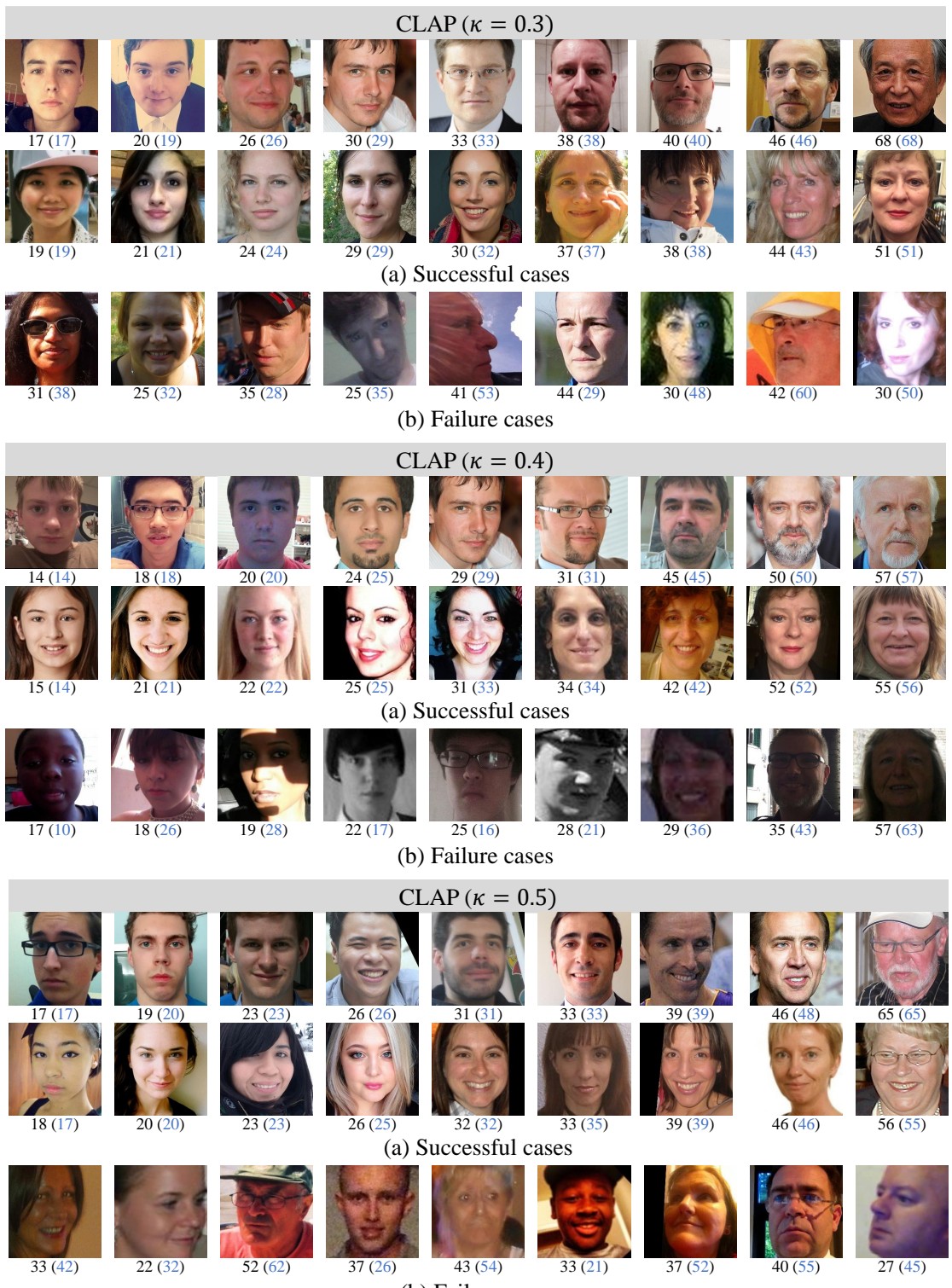

*Figure 16.* (a) Success and (b) failure cases of age estimation results on the CLAP2015 dataset. Under each image, the estimated ages are specified with the ground-truth in parentheses.

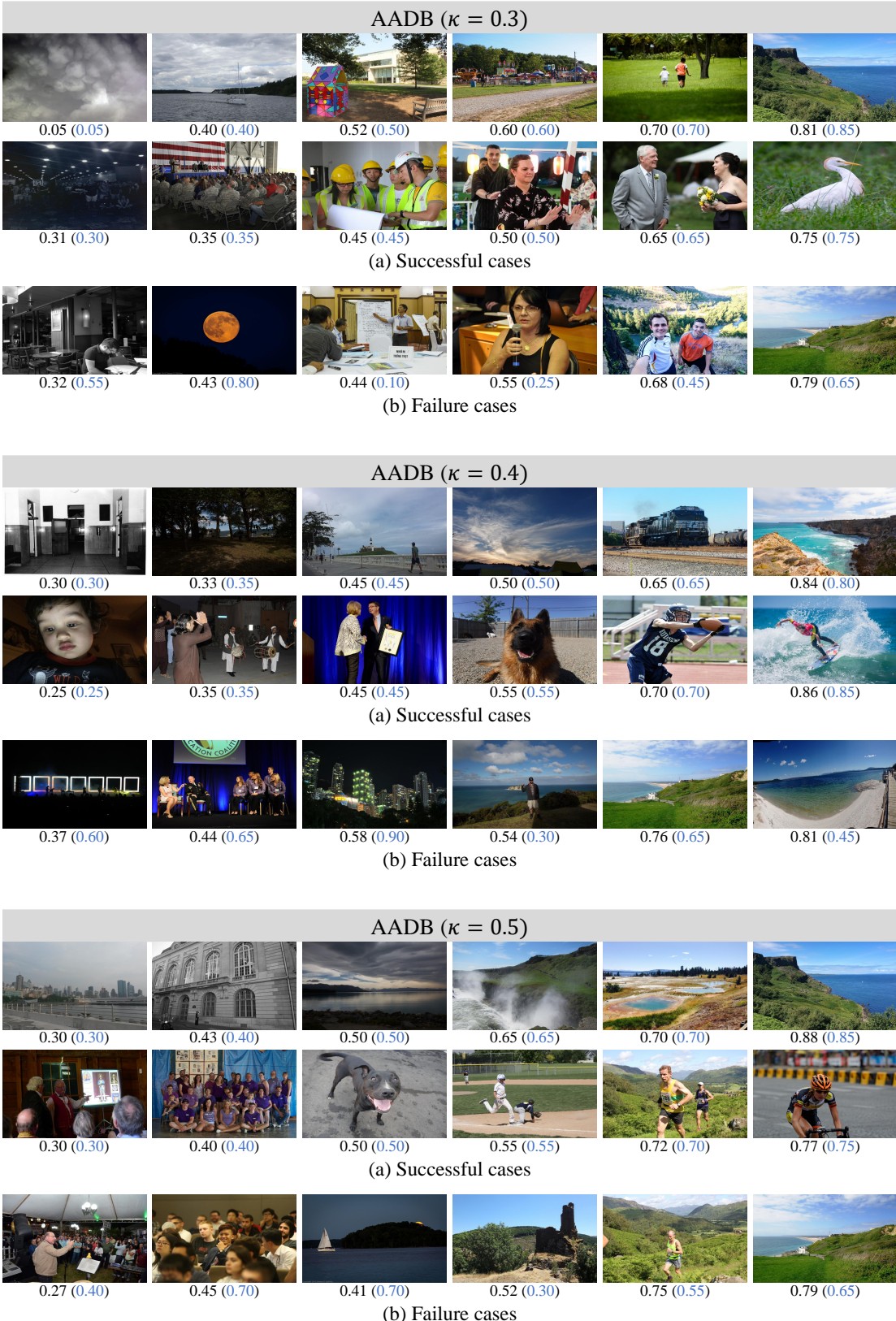

*Figure 17.* (a) Success and (b) failure cases of aesthetic score estimation results on the AADB dataset. Under each image, the estimated scores are specified with the ground-truth in parentheses.

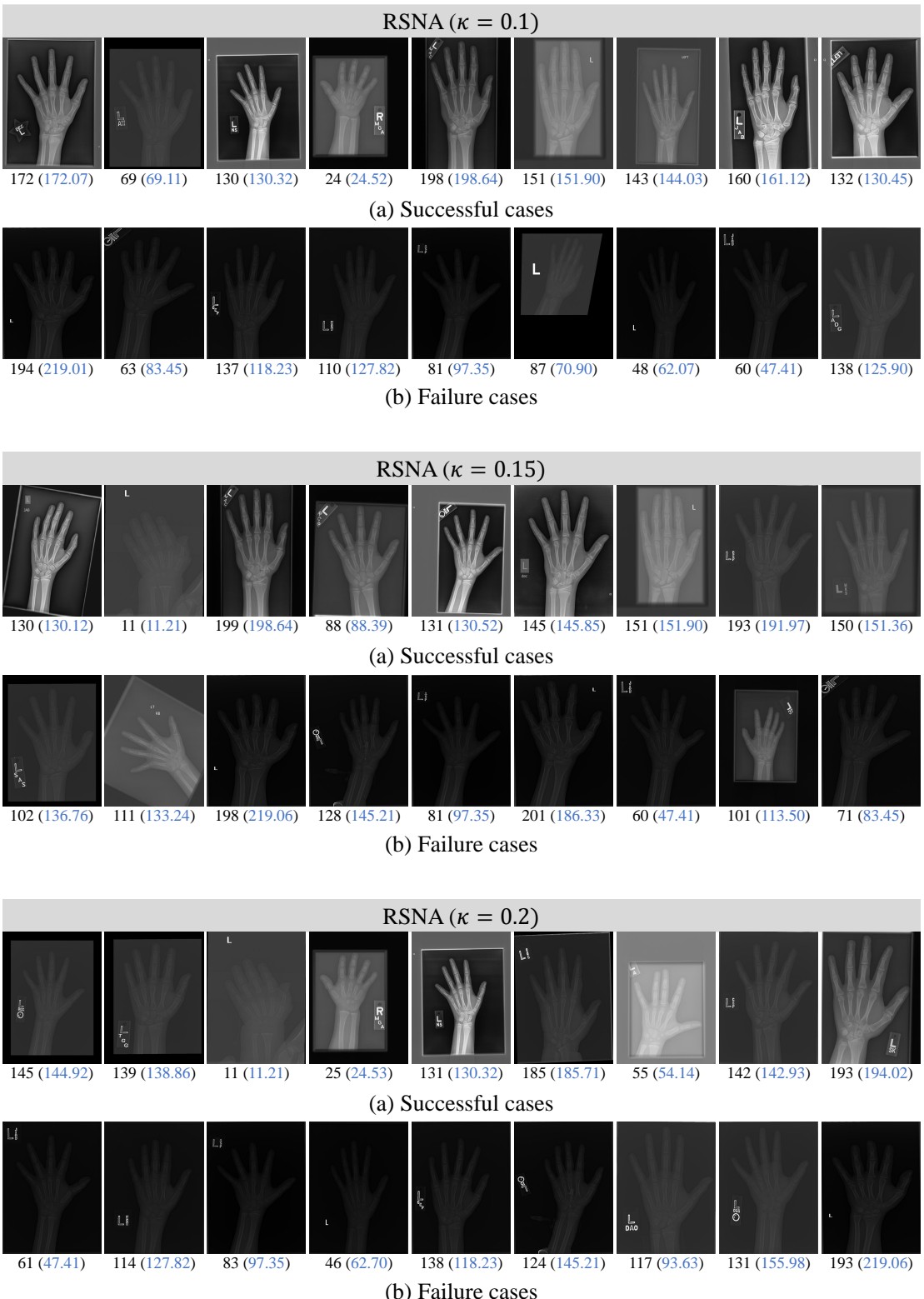

*Figure 18.* (a) Success and (b) failure cases of bone age assessment results on the RSNA dataset. Under each image, the estimated ages (in months) are specified with the ground-truth in parentheses.

## D.15. More Examples of Detected Outliers

Figures 19, 20, and 21 show examples of detected outliers on the MORPH II, CLAP2015, and AADB datasets, respectively.

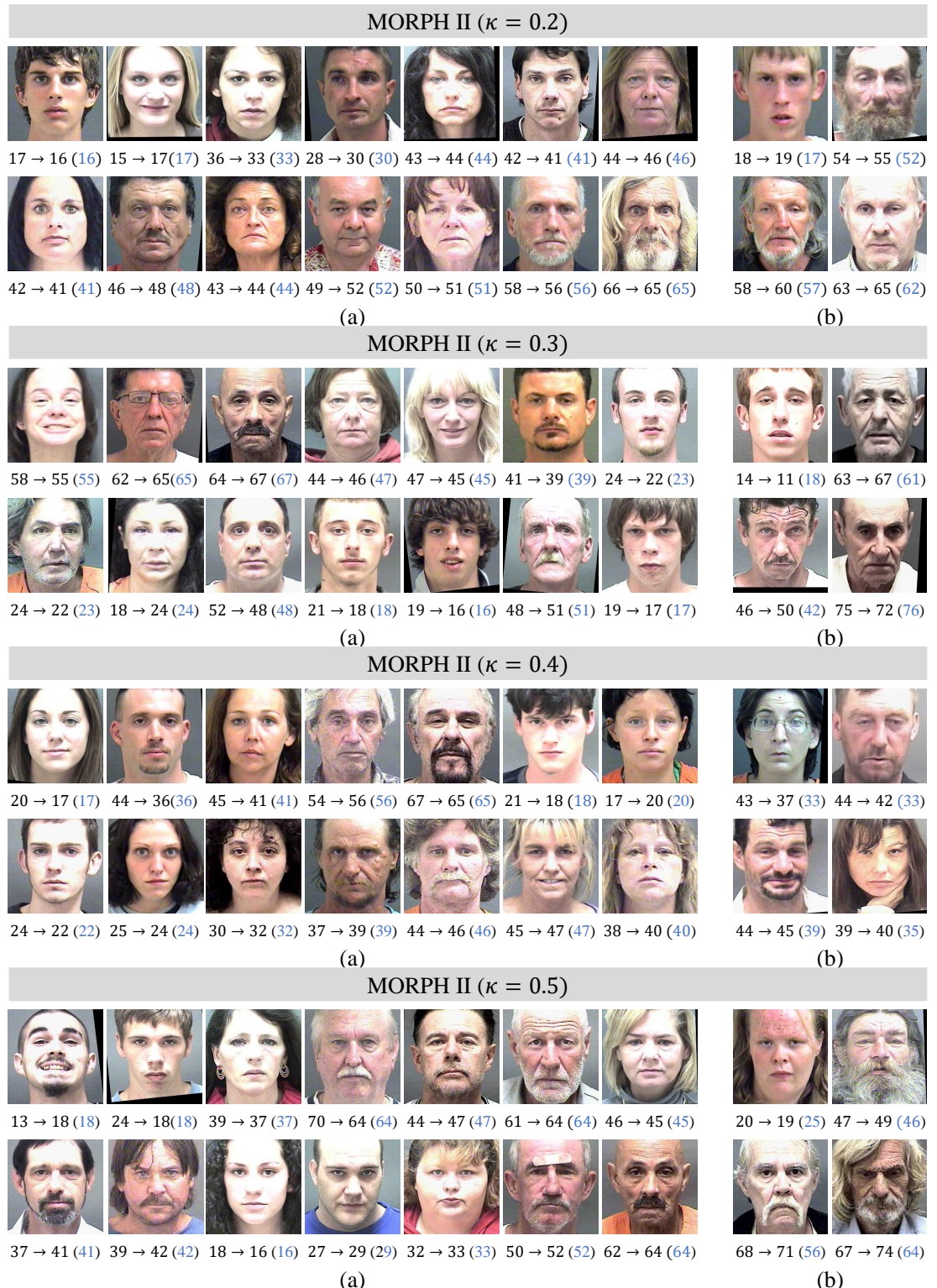

*Figure 19.* (a) Success and (b) failure cases of the label refinement on the MORPH II dataset. Under each image, the noisy, refined, and true ranks are specified: noisy → refined (true).

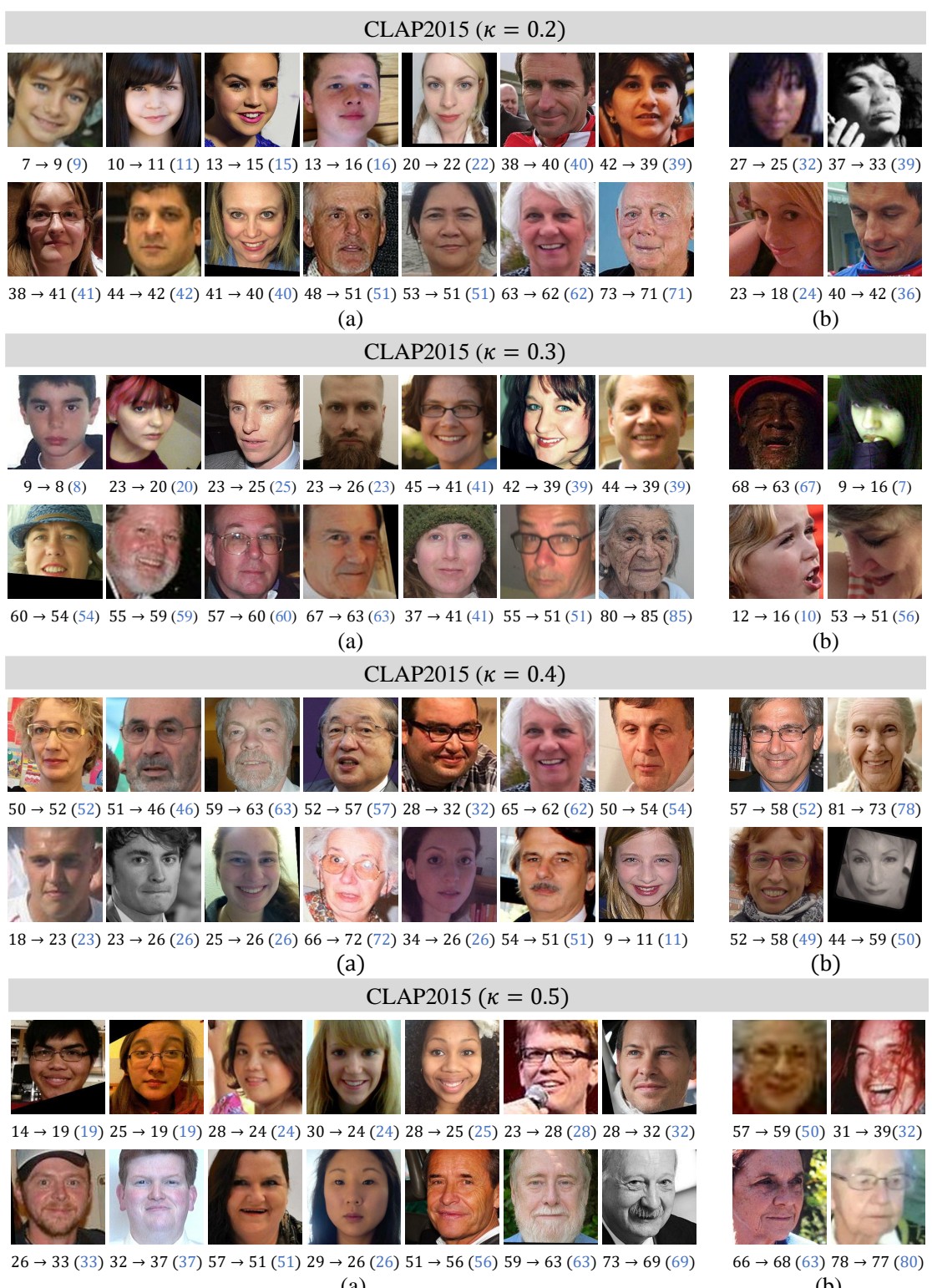

*Figure 20.* (a) Success and (b) failure cases of the label refinement on the CLAP2015 dataset. Under each image, the noisy, refined, and true ranks are specified: noisy → refined (true).

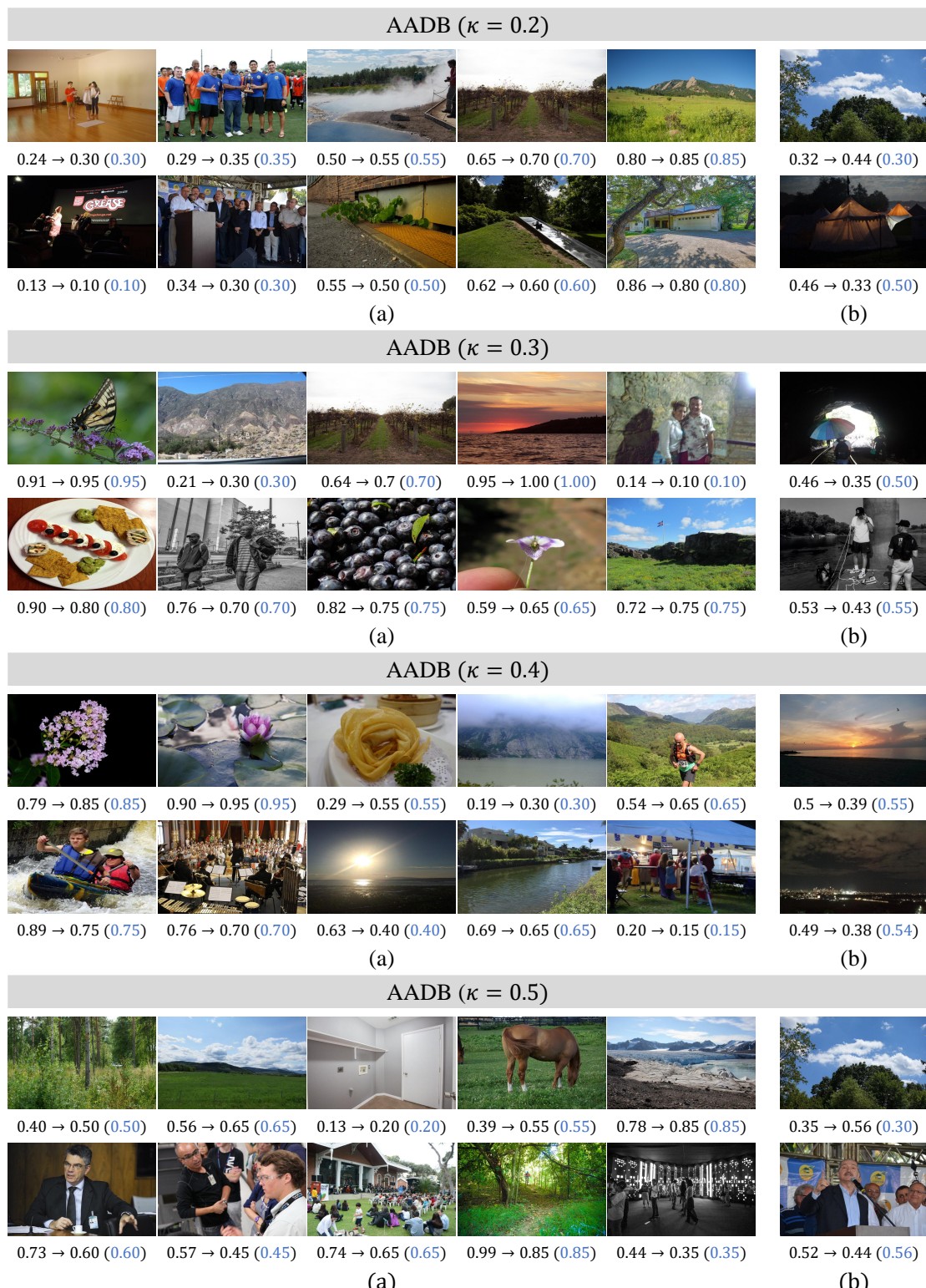

*Figure 21.* (a) Success and (b) failure cases of the label refinement on the AADB dataset. Under each image, the noisy, refined, and true ranks are specified: noisy → refined (true).

## E. Limitations

SOL has several limitations. First, it uses a fixed stochastic noise distribution to model ordinal label corruption. Although our experiments show robustness under various global and input-dependent noise settings, learning instance- or annotator-dependent noise distributions remains an important future direction. Second, the outlier detection and relabeling module is a practical enhancement rather than a direct consequence of the stochastic ordering formulation. Its benefit may therefore depend on the quality of the learned embedding space and the severity of label noise, making conservative relabeling preferable in practice. Finally, SOL involves several hyperparameters, such as $\sigma_{\text{test}}$, $T$, $\tau$, and $\beta$. While fixed default values work reliably in our experiments, fully automatic parameter selection is left for future work.

