# OpenReview forum: "Stochastic Order Learning: An Approach to Rank Estimation Using Noisy Data"
_ICML.cc/2026/Conference — ICML 2026 regular_

### Official Review · Reviewer_wTz4 · 2026-02-27

**Soundness:** 3
**Presentation:** 4
**Significance:** 2
**Originality:** 2
**Overall Recommendation:** 4
**Confidence:** 3

**Summary:**

This paper points out that in practical ranking estimation tasks, the correspondence between samples and ranking labels is typically stochastic due to the pervasive presence of noise. Addressing this characteristic, the authors propose SOL, which reformulates ranking estimation under noise as a stochastic ordering problem. They posit that a sample is associated with multiple adjacent ranking labels with varying probabilities, thereby capturing the structure of ranking label noise. From this perspective, SOL maps samples to an embedding space via an encoder and designs two complementary loss functions: a discriminative loss governs the similarity and dissimilarity between each sample and the centroid of its ranking label, and a stochastic order loss ensures global ranking consistency. Finally, the authors validate SOL's effectiveness across multiple benchmark datasets.

**Compliance With Llm Reviewing Policy:**

Affirmed.

**Final Justification:**

My concerns have been adequately addressed. I have raised my rating.

**Key Questions For Authors:**

- What are the advantages of SOL over existing LDL methods?
- Can SOL achieve better performance than state-of-the-art LDL algorithms in real-world datasets?

**Limitations:**

yes

**Strengths And Weaknesses:**

The paper is well-structured and clearly written, with excellent visual explanations. However, its novelty appears limited. Firstly, the core idea that the correspondence between samples and ranking labels is stochastic has been widely explored, and Label Distribution Learning (LDL) paradigm [1,2] has been established to address this problem. The paper does not clearly articulate its technical innovation compared to existing LDL approaches. Specifically, the utilization of a discrete Gaussian to model the uncertainty in the correspondence between samples and adjacent ranking labels is an idea already adopted in previous works [2,3,4]. Moreover, the proposed technique for handling Gaussian ranking labels does not demonstrate a clear advantage over methods like [2,3,4]; in my view, the commonly adopted KL divergence is sufficient to capture probabilistic ranking labels. Furthermore, the experimental section lacks comparisons with the state-of-the-art LDL methods, which is a significant omission for validating the claimed contributions.

[1] Xin Geng. Label Distribution Learning. IEEE Transactions on Knowledge and Data Engineering (TKDE), 2016, 28(7): 1734-1748.

[2] Xin Geng, Chao Yin, and Zhi-Hua Zhou. Facial Age Estimation by Learning from Label Distributions. IEEE Transactions on Pattern Analysis and Machine Intelligence (IEEE TPAMI), 2013, 35(10): 2401-2412.

[3] Bin-Bin Gao, Chao Xing, Chen-Wei Xie, Jianxin Wu, and Xin Geng. Deep Label Distribution Learning with Label Ambiguity. IEEE Transactions on Image Processing (IEEE TIP), 2017, 26(6): 2825-2838.

[4] Bin-Bin Gao, Hong-Yu Zhou, Jianxin Wu, and Xin Geng. Age Estimation Using Expectation of Label Distribution Learning. In: Proceedings of the International Joint Conference on Artificial Intelligence (IJCAI'18), Stockholm, Sweden, 2018, 712-718.

---

> ### Author Rebuttal · Authors · 2026-03-27
>
> We thank the reviewer for the thoughtful feedback and for pointing out the relevant label distribution learning (LDL) literature [1]-[4]. While SOL is related to this line of work, we respectfully disagree that it is subsumed by LDL.
>
> In the revised manuscript, we will explicitly discuss these works — including the general LDL formulation (Geng, 2016), the age-estimation framework (Geng et al., 2013), the ambiguity-aware extension (Gao et al., 2017), and the expectation-based method (Gao et al., 2018) — and clarify how SOL differs from them in both formulation and objective. Below, we explain these conceptual differences and provide additional empirical comparisons.
> ***
> > **Fundamental differences from LDL**
>
> SOL and label distribution learning (LDL) differ fundamentally in the problem formulation, the role of the Gaussian, and the learning objective.
>
> **(1) Problem formulation: distribution learning vs. stochastic ordering**
>
> * In LDL, a label distribution serves as the supervision target, and the model learns to match that distribution on a per-instance basis.
> * In SOL, the observed labels are treated as corrupted, and the goal is instead to recover the underlying ordinal structure under noise.
>
> In short:
>
> * LDL asks *“what label distribution should this instance match?”*
> * SOL asks *“what ordering structure is most consistent with the data despite corrupted labels?”*
>
> **(2) Role of the Gaussian: supervision target vs. noise model**
>
> While both frameworks use Gaussian functions, their roles are fundamentally different.
>
> * In LDL, the Gaussian defines the target label distribution to be fitted.
> * In SOL, the Gaussian defines a stochastic noise model ($p_s$) that weights geometric relations in the embedding space:
>   $D_h(x,r) = \sum_s p_s d^2(h_x, \mu_{r+s}).$
>
> Thus, in SOL, the Gaussian is not used to define the supervision target; rather, it is used to model how corrupted ordinal labels relate to neighboring ranks.
>
> **(3) Learning objective: per-instance matching vs. relational ordering**
>
> * LDL typically optimizes independent per-instance distribution matching (e.g., via KL divergence).
> * SOL instead learns an embedding space through stochastic dissimilarity and a stochastic order loss, which together enforce ordering consistency under uncertainty.
>
> Thus, unlike standard LDL frameworks, SOL is not based on per-instance target matching alone, but on learning ordinal geometry and ordering-aware representations under noisy ordinal labels.
>
> &nbsp;
>
> > **Performance comparison with LDL methods**
>
> We conducted additional experiments comparing SOL with state-of-the-art LDL methods (DLDL-v2, Uni-Con, DHRL) under multiple noise settings. SOL consistently outperforms all LDL baselines across noise types and levels. This supports our claim that the difference is not merely terminological, but lies in how uncertainty is modeled. LDL captures ambiguity around the observed label, whereas SOL explicitly models label corruption over neighboring ranks. The advantage is especially clear at higher noise levels (e.g., $\kappa=0.4$). These results will be included in the revised manuscript.
>
> ||Gaussian $\kappa=0.2$||Gaussian $\kappa=0.3$||Gaussian $\kappa=0.4$||Laplacian $\kappa=0.3$||Uniform $\kappa=0.3$||Skewed $\kappa=0.3$||
> |-|-|-|-|-|-|-|-|-|-|-|-|-|
> |Algorithm|MAE$(\downarrow)$|CS$(\uparrow)$|MAE$(\downarrow)$|CS$(\uparrow)$| MAE$(\downarrow)$|CS$(\uparrow)$|MAE$(\downarrow)$|CS$(\uparrow)$|MAE$(\downarrow)$|CS$(\uparrow)$|MAE$(\downarrow)$|CS$(\uparrow)$|
> |DLDL-v2 [A]|2.753|84.34|2.882|84.79|3.206|77.60|3.199|78.42|2.789|83.79|3.364|77.69|
> |Uni-Con [B]|2.737|85.43|2.835|85.06|3.213|78.87|3.247|78.05|2.714|85.52|3.303|79.51||
> |DHRL [C]|2.609|70.49|2.737|85.34|2.870|82.60|2.992|80.41|2.617|86.70|3.305|78.05|
> | SOL |**2.489**|**91.35**|**2.663**|**89.62**|**2.826**|**87.70**|**2.986**|**85.88**| **2.499**|**90.89**|**3.296**|**83.15**|
>
> [A] Gao et al., "Age estimation using expectation of label distribution learning," IJCAI 2018.
>
> [B] Li et al., "Unimodal-concentrated loss: Fully adaptive label distribution learning for ordinal regression," CVPR 2022.
>
> [C] Suzuki et al., "Distribution highlighted reference-based label distribution learning for facial age estimation," WACV 2026.
>
> ***
> > **Summary**
>
> We have addressed the reviewer's concerns through the following improvements.
>
> * clarification of the distinctions between SOL and LDL
> * additional comparisons with state-of-the-art LDL methods
>
> We sincerely appreciate the reviewer’s careful reading and the important question regarding the relationship between SOL and LDL. This comment prompted us to clarify the distinction more explicitly, which has strengthened the paper. The additional experiments further support this distinction, showing that SOL consistently outperforms state-of-the-art LDL methods across all noise types and levels. We hope these clarifications and comparisons adequately address the reviewer’s concerns, and we would be happy to discuss any remaining points.

---

> > ### Author Rebuttal · Reviewer_wTz4 · 2026-04-01
> >
> > My concerns have been adequately addressed.

---

### Official Review · Reviewer_DAcN · 2026-03-10

**Soundness:** 3
**Presentation:** 4
**Significance:** 4
**Originality:** 3
**Overall Recommendation:** 5
**Confidence:** 4

**Summary:**

This paper proposes a novel framework for the ordinal regression problem with noisy data. The approach aims to find an embedding space $h(\cdot)$ using a constructed loss function consider two objectives: (1) the instance in $h(\cdot)$ should be close to the center of its class, and (2) two instances should be in correct order.

**Compliance With Llm Reviewing Policy:**

Affirmed.

**Final Justification:**

Thank you for the response. I will keep my positive score.

**Key Questions For Authors:**

- From Algorithm 1, the procedure should converge, since it relabels the outliers. However, how to ensure the converged encoder $h$ is sufficiently good? Is it possible that the encoder is not good in the "Network Training" step, and then the algorithm relabels all the correct data with wrong labels in the "Relabeling" step? Since all parameters have their physical meaning in loss functions, how to select the parameters or models to avoid the bad encoder $h$?

- Should the parameter $\sigma$ in Eq. (2), $T$ in Eq. (8), $\tau$ in Eq. (11), and $\beta$ in Eq. (19) be part of the input in the algorithm?

- It seems that the parameter $\sigma$ in Eq. (2) is related to the total rank and noisy level. In Table 7, the exact values of $\sigma$ in CLAP2015 are 4.941 when $k=0.4$ and 6.117 when $k=0.5$. From Figure 8, the performance reduces when $\sigma_{test} \ge 4$. Should the performance be good when the exact values of $\sigma$ are equal to $\sigma_{test}$?

- About the weakness mentioned above, is there any explanation? What is the threshold $\delta$ (right-hand side of lines 190-203) when $T=1, \sigma_{test} = 1$ in the experiment?

**Limitations:**

The limitation is not discussed in the paper.

**Strengths And Weaknesses:**

Strengths:
- The proposed framework considers both the label and the relation between instances, which can be robust to label noise.

- The idea of the method is straightforward to understand and is reasonable.

- The framework can be applied for both ordinal regression and outlier detection.

Weaknesses:
- There are a lot of ranks in each dataset, such as 61 ranks in MORPH II and 82 ranks in CLAP2015. Ideally, the $T$ in Eq. (8) should consider more ranks around the given label. However, the $T$ is set to be 1 in experimental comparison ($T=2,3$ in Table 10), meaning that it does not need to take care of the ranks around the given label. This fact can potentially imply that the $T$ in Eq. (8) is redundant.

---

> ### Author Rebuttal · Authors · 2026-03-27
>
> We sincerely thank the reviewer for their careful reading and positive assessment. We address each question below.
> ***
> > **Convergence and bad encoder risk**
>
> We agree that the interaction between encoder quality and relabeling is critical. Our design incorporates three key safeguards to ensure stable convergence and prevent error accumulation.
>
> - **Conservative relabeling:** The threshold $\beta$ in Eq. (19) strictly limits the scope of relabeling. With $\beta = 0.9$, at most 10% of samples per rank are updated in each iteration, preventing large-scale label corruption.
> - **Stable update magnitude:** Eq. (20) utilizes the global mean absolute deviation rather than instance-specific updates, which suppresses noise from individual prediction errors and avoids oscillatory behavior.
> - **Strong initialization and gradual refinement:** The encoder is initialized with ImageNet-pretrained weights and refined before each relabeling step, ensuring that relabeling is applied only after a meaningful ordinal structure has been learned.
>
> Empirically, Table 20 and Figure 10 demonstrate that both the average noise level and the number of extreme-noise instances decrease after refinement, confirming stable convergence. To avoid unstable encoders in practice, we recommend setting $\beta \geq 0.85$ to limit aggressive relabeling and performing the relabeling step only after sufficient pre-training. We will include these guidelines in the revised manuscript.
>
> &nbsp;
>
> > **Algorithm parameters**
>
> While the parameters $\sigma$, $T$, $\tau$, and $\beta$ are detailed in the main text, we agree that explicitly listing them as inputs in Algorithm 1 will enhance clarity. We will revise the algorithm block accordingly.
>
> &nbsp;
>
> > **$\sigma$ (noise generation) vs. $\sigma_{\text{test}}$ (algorithm parameter)**
>
> We clarify that $\sigma_{\text{test}}$ is not intended to match the true noise level $\sigma$. Instead, it effectively acts as a temperature parameter controlling the sharpness of the stochastic ordering distribution. When $\sigma_{\text{test}}$ is set to excessively large values (e.g., matching a high true noise scale such as 4.9 or 6.1 in CLAP2015), the distribution becomes overly flat, assigning nearly uniform weights across rank centroids. As shown in Figure 8, this weakens rank discrimination and consequently degrades performance. Therefore, optimal performance is achieved not by matching the true noise scale, but by maintaining sufficient contrast in the ordering probabilities. This explains why a fixed, moderate value (e.g., $\sigma_{\text{test}}=1$) works consistently well in practice across various datasets.
>
> &nbsp;
>
> > **Choice of $T$ in Eq. (8) and threshold $\delta$**
>
> We address this by clarifying two points: (1) why setting $T=1$ is not redundant, and (2) how the concrete value of $\delta$ is determined when $T=1$.
>
> - **$T=1$ is not redundant:**
> Setting $T=1$ does not make the loss consider only the exact label. As shown in Eq. (8), the formulation inherently incorporates all offsets $s$ through the stochastic weights $p_s$. Therefore, neighboring ranks are deeply integrated into the optimization process via $p_s$, even when $T=1$.
>
> - **Concrete value of $\delta$ when $T=1$:**
> The threshold $\delta$ is defined by the sign change of $\alpha_s$. Specifically, $\alpha_s > 0$ for $|s| < \delta$ and $\alpha_s < 0$ otherwise. For $T=1$, the coefficient becomes $\alpha_s = 2p_s - p_{s-1} - p_{s+1}$.
> Under our experimental setting of $\sigma_{\text{test}}=1$, the Gaussian weights decay rapidly, $p_0 : p_1 : p_2 \approx 1.000 : 0.607 : 0.135$.
> This rapid decay yields $\alpha_0 > 0$, $\alpha_1 > 0$, and $\alpha_2 < 0$. Consequently, the threshold is exactly $\delta = 2$.
>
> &nbsp;
>
> > **Limitations**
>
> Following your suggestion, we will add a dedicated limitations section in the revised manuscript to discuss the scope and bounds of our approach.
>
>
> ***
> > **Summary**
>
> We have addressed the reviewer’s questions through:
>
> - clarification of the convergence behavior and algorithmic safeguards against error accumulation
>
> - additional discussion on the roles of key parameters and practical selection guidelines
>
> - clarification of the functional role of $\sigma_{\text{test}}$ as a temperature parameter
>
> - a mathematical explanation for the choice of $T$ in Eq. (8) and the resulting threshold $\delta$
>
> We sincerely appreciate the reviewer’s insightful feedback, which has significantly helped improve both the clarity and completeness of our paper.

---

> > ### Author Rebuttal · Reviewer_DAcN · 2026-04-02
> >
> > Thank you for the response. I will keep my positive score.

---

### Official Review · Reviewer_44jP · 2026-03-13

**Soundness:** 3
**Presentation:** 3
**Significance:** 3
**Originality:** 3
**Overall Recommendation:** 4
**Confidence:** 4

**Summary:**

This paper studies ordinal or rank estimation under noisy labels and argues that, once ordinal annotations are corrupted, the relationship between an instance and its rank should no longer be modeled as deterministic. Instead, the paper formulates the problem as stochastic ordering. Based on this view, the authors propose SOL, which learns an embedding space using a discriminative loss and a stochastic order loss, and can optionally refine training labels through an outlier-detection and relabeling module. The empirical evaluation spans facial age estimation, apparent age estimation, aesthetic score regression, bone-age assessment, and a WMT2020 text-regression task under several synthetic noise families and a real-noise setting. The authors strive to outline an important concept: ordinal label noise should be modeled as stochastic uncertainty rather than deterministic corruption. The article's primary contribution pertains to a stochastic order learning framework that associates each instance with multiple plausible ranks and learns an embedding space under noisy ordinal supervision.

**Compliance With Llm Reviewing Policy:**

Affirmed.

**Final Justification:**

I maintain my original rating of 4: Weak Accept.
The rebuttal did not change my assessment. While the technical approach is reasonable, my concerns regarding limited contribution and unclear writing remain. Borderline case.

**Key Questions For Authors:**

1. At test time, the method uses a fixed sigma_test to compute the probabilities in Eq. (2), but in deployment one often has neither a clean validation set nor prior knowledge of the noise level. Could the authors explain more clearly how sigma_test should be selected without extra supervision? A response with a default strategy, an unsupervised estimation method, or stronger robustness evidence would make the method more practically compelling.
2. The derivation is built around a symmetric discrete-Gaussian noise assumption. While the paper evaluates Laplacian, uniform, and skewed synthetic noise, these are still global distributional perturbations. Could the authors discuss or test what happens when the noise level depends on the input content, rank range, or sample difficulty? This would help define the scope of applicability more clearly.
3. Please report mean plus or minus standard deviation over multiple random seeds for at least representative datasets and noise settings. Several gains over strong baselines are relatively modest in absolute terms, so seed-level stability would strengthen the empirical case substantially.
4. The outlier-detection and relabeling module appears helpful overall, but the magnitude of its gain varies across datasets and sometimes involves trade-offs. Could the authors clarify more explicitly when this module is most useful, when it may be unnecessary, and whether it should be viewed as an optional component rather than as a core part of the main theoretical contribution?

**Limitations:**

Partly yes. The impact statement appropriately notes potential bias when the method is used on facial datasets, and the appendices provide useful sensitivity and complexity analyses. However, I still encourage the authors to discuss three technical limitations more explicitly in the main paper: (i) the dependence on a fixed parametric noise model, (ii) the heuristic nature of the label-refinement module, and (iii) the lack of multi-seed statistics for assessing result stability. Beyond the already stated dataset-bias concern, I do not currently have an additional ethics flag.

**Strengths And Weaknesses:**

Below I assess the paper separately along the dimensions of soundness, presentation, significance, and originality.

Soundness
The method is internally coherent: the paper moves from noisy ordinal labels to stochastic dissimilarity, then to the discriminative loss, stochastic order loss, centroid updates, and the inference rule in a logically consistent way. The empirical section is also the strongest part of the paper, with broad coverage across datasets, noise types, a real-noise NLP setting, ablations, sensitivity analysis, and complexity analysis. My main reservations on soundness are threefold. First, the probabilistic formulation still fundamentally assumes a fixed, symmetric discrete-Gaussian noise model. Although the experiments show robustness under several alternative synthetic noise types, the method itself is not yet analyzed systematically for instance-dependent or more structured annotation noise. Second, the outlier-detection and relabeling module in Eqs. (19)-(21) is heuristic rather than naturally derived from the same framework. Third, most main tables report single numbers without mean and standard deviation over multiple random seeds, so the stability of some gains remains unclear.

Presentation
The paper is generally well written. The motivation is clear, Figures 2 and 3 are useful, and the experiments are organized well. That said, Section 3 is still notation-heavy, especially around the stochastic order loss, the ordering probabilities, and the relationship between the core SOL formulation and the relabeling module. The main text would be improved by more explicitly separating the contribution of the stochastic-order formulation from the additional gain provided by the optional label-refinement stage, especially because the appendices already provide partial disentanglement through ablations.

Significance
I view the problem as important. Ordinal noise has structure, and simply reusing noisy-classification methods is not a satisfactory solution. The paper articulates this well and demonstrates potential applicability across multiple vision tasks and one NLP task, which gives the work clear value for noisy ordinal prediction. Even if the paper may not yet provide the final word on the theory of noisy ordinal learning, it offers a constructive framework that future work is likely to build on.

Originality
The most original aspect of the paper is the reformulation itself: noisy rank estimation is cast as stochastic ordering rather than ordinary ordinal regression plus a robustness add-on. This perspective is fresh, and the loss design is tied to it in a reasonably principled way. By contrast, the outlier-relabeling module feels more like a useful engineering addition than the core conceptual contribution. Overall, I consider the central formulation original and worthwhile, while viewing the refinement module as an auxiliary enhancement.

---

> ### Author Rebuttal · Authors · 2026-03-27
>
> We thank the reviewer for the detailed and thoughtful feedback. We address the four questions below and will incorporate the clarifications into the revised manuscript.
>
> ***
>
> > **Fixed $\sigma_{\text{test}}$**
>
> We agree that matching $\sigma_{\text{test}}$ to the underlying noise scale is intuitively reasonable. However, in our formulation,  $\sigma_{\text{test}}$ serves as a *weighting parameter in the stochastic dissimilarity*, rather than a direct estimate of the true noise level. As a result, performance does not rely on exact matching.
>
> - **Robustness:** As shown in Appendix D.2, performance is largely insensitive to $\sigma_{\text{test}}$ within a reasonable range.
> - **Extreme values:** Only excessively large values degrade performance, as they overly flatten the distribution and weaken rank discrimination.
> - **Default strategy:** We used a fixed default (e.g., $\sigma_{\text{test}}=1$) across all experiments. The consistently strong results across diverse datasets and noise settings demonstrate that this default works reliably without validation data or prior knowledge of the noise level.
>
> &nbsp;
>
> > **Input-dependent noise**
>
> We thank the reviewer for this suggestion. To assess the robustness of SOL beyond global perturbation models, we additionally evaluate an input-dependent noise setting motivated by a common observation in age estimation: labels for very young or elderly subjects are relatively easier to estimate, whereas middle-aged samples are often more ambiguous and thus more prone to annotation noise. Specifically, we define a rank-dependent modulation
> $f_{\text{rank}}(r) = \exp\left(-\frac{(r-r_{\text{peak}})^2}{2w^2}\right),$
> with $r_{\text{peak}}=40$ and $w=15$, and set the instance-level noise as
> $\sigma(x)=\kappa \cdot \sigma_{\mathcal{X}} \cdot (1+f_{\text{rank}}(r_x)),$
> which yields higher noise around ambiguous age ranges.
>
> On the MORPH II dataset with $\kappa=0.3$, SOL continues to outperform the strong GOL baseline on both MAE and CS. These results indicate that SOL remains robust under input-dependent noise, beyond a single global noise model.
>
>
> |Method|MAE($\downarrow$)|CS($\uparrow$)|
> |-|-|-|
> |GOL|2.724|88.25|
> |SOL|**2.702**|**88.71**|
>
> &nbsp;
>
> > **Multi-seed statistics**
>
> We report mean $\pm$ std performance over five random seeds on MORPH II (Gaussian noise at $\kappa=0.2$), in comparison with state-of-the-art methods. The results show that SOL maintains strong performance with low variance, demonstrating that the gains are consistent and not due to favorable random seeds. We will include these statistics in the revised manuscript. Furthermore, we are currently conducting additional multi-seed experiments under other noise settings to further verify the stability of our method, and will provide those results as they become available.
>
>
> | Algorithm| MAE$(\downarrow)$ | CS$(\uparrow)$ |
> |-|-|-|
> |MWR|2.574$\pm$0.019|89.816$\pm$0.505|
> |GOL|2.520$\pm$0.023|90.802$\pm$0.440|
> | SOL |**2.497$\pm$0.006**|**90.984$\pm$0.456**|
>
> &nbsp;
>
> > **Outlier-detection and relabeling module**
>
> The outlier-detection and relabeling module is primarily beneficial in high-noise regimes.
>
> - **When useful:** It provides more noticeable gains in high-noise, large-scale settings (e.g., RSNA), where label inconsistency is substantial and correcting mislabeled samples can significantly improve training.
> - **When less impactful:** The improvement is modest on smaller or relatively clean datasets (e.g., CLAP2015), where label noise is limited and fewer samples require correction.
>
> Given this behavior, the module is best viewed as an optional practical enhancement for challenging noise regimes rather than a necessary component of SOL.
>
> &nbsp;
> ***
> > **Summary**
>
> We have addressed the reviewer’s concerns through:
>
> * clarification of the role and robustness of $\sigma_{\text{test}}$
> * additional experiments and discussion on input-dependent noise
> * inclusion of multi-seed statistics
> * clarification of the role of the relabeling module
>
> We appreciate the reviewer’s insightful comments, which helped improve the clarity and positioning of the paper.

---

> > ### Author Rebuttal · Reviewer_44jP · 2026-04-07
> >
> > The authors have comprehensively addressed all four questions.

---

### Official Review · Reviewer_bpFS · 2026-03-14

**Soundness:** 3
**Presentation:** 3
**Significance:** 3
**Originality:** 3
**Overall Recommendation:** 5
**Confidence:** 5

**Summary:**

This paper addresses the challenge of rank estimation in the presence of label noise. The authors argue that ordinal annotations often carry structured uncertainty that traditional deterministic labeling fails to capture. To address this, the paper reformulates the task as a stochastic ordering problem, where each instance is associated with multiple plausible ranks rather than a single fixed label. The proposed Stochastic Order Learning (SOL) framework aims to capture this uncertainty within a learned embedding space to improve robustness against label corruption.

**Compliance With Llm Reviewing Policy:**

Affirmed.

**Final Justification:**

My previous concerns have been adequately addressed, and I recommend to accept this paper.

**Key Questions For Authors:**

See the weaknesses above.

**Limitations:**

yes

**Strengths And Weaknesses:**

**Strengths:**
1. The authors proposed an effective method to solve the specific problem of structured ordinal uncertainty identified in this paper.
2. The paper is visually well-presented by combining illustrations with texts and easy to follow.

**Weaknesses:**
1. The experimental section primarily compares the proposed method against older baselines. The absence of comparisons with state-of-the-art methods from 2024 and 2025 weakens the claim of the proposed method’s competitiveness.

---

> ### Author Rebuttal · Authors · 2026-03-27
>
> We sincerely thank the reviewer for the positive feedback and for pointing out the need for comparisons with more recent methods.
>
> ***
>
> > **More comparisons**
>
> Following your suggestion, we have conducted additional experiments comparing SOL with recent state-of-the-art ordinal regression and rank estimation methods: ConR (ICLR 2024), CLOC (CVPR 2025), and DHRL (WACV 2026). Our method consistently outperforms these recent baselines across all settings. We will include these new comparisons and references in the revised manuscript.
>
> ||Gaussian $\kappa=0.2$||Gaussian $\kappa=0.3$||Gaussian $\kappa=0.4$||Laplacian $\kappa=0.3$||Uniform $\kappa=0.3$||Skewed $\kappa=0.3$||
> |-|-|-|-|-|-|-|-|-|-|-|-|-|
> | Algorithm| MAE$(\downarrow)$ | CS$(\uparrow)$ | MAE$(\downarrow)$ | CS$(\uparrow)$| MAE$(\downarrow)$ | CS$(\uparrow)$ | MAE$(\downarrow)$ | CS$(\uparrow)$|MAE$(\downarrow)$ | CS$(\uparrow)$ |MAE$(\downarrow)$|CS$(\uparrow)$|
> | ConR [1]|3.296|77.87|3.382|77.60|3.924|70.49|3.486|75.23|3.168|80.42|3.610|74.86|
> | CLOC [2]|4.461|67.58|5.966|52.64|6.232|50.27|8.018|38.43|3.558|79.87|3.502|79.14|
> |DHRL [3]|2.609|70.49|2.737|85.34|2.870|82.60|2.992|80.41|2.617|86.70|3.305|78.05|
> | SOL |**2.489**|**91.35**|**2.663**|**89.62**|**2.826**|**87.70**|**2.986**|**85.88**| **2.499**|**90.89**|**3.296**|**83.15**|
>
> [1] Keramati et al., "ConR: Contrastive regularizer for deep imbalanced regression," ICLR 2024.
>
> [2] Pitawela et al., "CLOC: Contrastive learning for ordinal classification with multi-margin N-pair loss," CVPR 2025.
>
> [3] Suzuki et al., "Distribution highlighted reference-based label distribution learning for facial age estimation," WACV 2026.
>
> ***
>
> We hope these additional comparisons adequately address your concern.

---

> > ### Author Rebuttal · Reviewer_bpFS · 2026-04-02
> >
> > My previous concerns have been adequately addressed, and I will accordingly raise my score.

---

### Decision · Program_Chairs · 2026-04-30

**Decision:**

Accept (regular)

**Comment:**

All four reviews of this work [bpFS,44jP,DAcN,wTz4] leaned towards acceptance, with two Accept recommendations and two Weak Accept recommendations.

The reviewers appreciated several positive aspects of the work:

+ The problem was considered important [44jP]
+ The idea of the method was considered straightforward and reasonable [DAcN]
+ Having an effective method to solve structured ordinal uncertainty was appreciated [bpFS]
+ Considering both the label and the relation between instances was appreciated [DAcN]
+ The reformulation where noisy rank estimation is cast as stochastic ordering was considered original and fresh [44jP]
+ The method was considered internally coherent [44jP]
+ Applicability to both ordinal regression and outlier detection was appreciated [DAcN]
+ The paper was considered well written [44jP,wTz4] and easy to follow and visually well-presented [bpFS,wTz4]

However, several concerns were also brought up:

- Novelty was considered limited [wTz4]
- Lack of discussion of novelty compared to existing LDL approaches was criticized [wTz4]; authors provided some new discussion of methodological differences.
- Lack of comparison to newer baselines was criticized [bpFS]; similarly, lack of experimental comparison with state of the art LDL methods was criticized [wTz4]; authors responded and added some new baselines.
- Assumption of a fixed, symmetric discrete-Gaussian noise model and lack of analysis for instance-dependent or structured annotation noise was criticized [44jP]; authors provided an additional evaluation of an input-dependnt noise setting.
- The outlier-detection and relabeling module was considered heuristic [44jP]
- Considering more ranks around the given label than in current experiments was desired [DAcN]; authors argued setting T=1 inherently incorporates all offsets through stochastic weights.
- It was considered unclear when the outlier-detection and relabeling module is most helpful [44jP]; authors provided some discussion.
- How to ensure the converged encoder is sufficiently good was considered unclear [DAcN]; authors argued their design incorporates safeguards.
- Discussion of seemingly weak performance with one parameter setting was desired [DAcN]; authors provided some discussion.
- Lack of standard deviations over random seeds in results was criticized [44jP]; authors provided some statistics.
- Some areas of writing were considered notation-heavy [44jP]
- Selection of sigma_test was considered unclear [44jP]
- Lack of a clear advantage of the technique for handling Gaussian ranking over existing methods was criticized [wTz4]
- Discussion of limitations was desired [DAcN] ;authors promised to add one.

The authors responded to the concerns; in particular, they included argumentation and additional comparisons to new baselines which seemed to satisfy [bpFS], addressed the issues raised by [44jP] and [DAcN] to the reviewers' satisfaction, and compared to LDL methods in a response to [wTz4] which seemed to satisfy the concerns of [wTz4].

Overall, the work seems to have clear interest. While there were strong concerns especially regarding novelty and sufficient comparison to state of the art, it seems authors were able to sufficiently address them in their responses. Therefore, I believe the work could be accepted to ICML.